# Antigen recognition reinforces regulatory T cell mediated *Leishmania major* persistence

Romaniya Zayats [1], Zhirong Mou[1], Atta Yazdanpanah[1], Gaurav Gupta[1], Paul Lopez[1], Deesha Nayar[1], Wan H. Koh [1], Jude E. Uzonna[1,2] & Thomas T. Murooka [1,2]

Cutaneous *Leishmania major* infection elicits a rapid T cell response that is insufficient to clear residually infected cells, possibly due to the accumulation of regulatory T cells in healed skin. Here, we used *Leishmania*-specific TCR transgenic mice as a sensitive tool to characterize parasite-specific effector and immunosuppressive responses in vivo using two-photon microscopy. We show that *Leishmania*-specific Tregs displayed higher suppressive activity compared to polyclonal Tregs, that was mediated through IL-10 and not through disrupting cell-cell contacts or antigen presentation. In vivo expansion of endogenous *Leishmania*-specific Tregs resulted in disease reactivation that was also IL-10 dependent. Interestingly, lack of Treg expansion that recognized the immunodominant *Leishmania* peptide PEPCK was sufficient to restore robust effector Th1 responses and resulted in parasite control exclusively in male hosts. Our data suggest a stochastic model of *Leishmania major* persistence in skin, where cellular factors that control parasite numbers are counterbalanced by *Leishmania*-specific Tregs that facilitate parasite persistence.

Leishmaniasis is a neglected tropical disease caused by the protozoan parasites of the genus *Leishmania*. Infected individuals mount a strong Th1 cell response, characterized by robust delayed-type hypersensitivity reactions and IFNγ production that enables killing of most intracellular parasites in host macrophages and dendritic cells[1–3]. Much emphasis has been placed on generation of acute anti-*Leishmania* immunity, whereas the establishment and maintenance of long-lived parasitic reservoirs are less well understood, mainly due to the complexity and ongoing immune evolution at the host-parasite interface[4,5]. However during leishmaniasis, a small population ($10^2$–$10^4$) of parasites continue to persist in healed skin despite the generation of robust Th1 immunity[6–8], where tissue reservoirs maintain concomitant immunity against *Leishmania* reinfections[9–11]. Importantly, such long-lived parasitic reservoirs pose a significant risk of disease reactivation in infected individuals[12,13].

Two non-mutually exclusive models have been proposed to explain why sterile cure is not achieved. The presence of "sanctuary sites" may allow specific subsets of susceptible cells to facilitate long-term *Leishmania* survival in skin, such as alternatively-activated macrophages lacking iNOS expression necessary for parasite killing[14]. However, more recent characterization of persistent *L. major* parasites within iNOS+ host cells argues against the existence of sanctuary cells[15], although continual low influx of inflammatory monocytes could contribute to host reservoir longevity[16]. Another explanation is the accumulation of regulatory T cells (Tregs) and immunosuppressive cytokine IL-10 that function to dampen host immunity to help maintain parasite persistence. Depletion of CD4+CD25+ Tregs or the blockade of IL-10 signaling results in sterilizing cure in *Leishmania*-infected mice[9,17], whereas treatment of healed mice with immunosuppressive drugs or blockade of IFNγ signaling resulted in a rapid increase in parasite

[1]Department of Immunology, Rady Faculty of Health Sciences, University of Manitoba, Winnipeg, MB, Canada. [2]Department of Medical Microbiology and Infectious Diseases, Rady Faculty of Health Sciences, University of Manitoba, Winnipeg, MB, Canada. e-mail: Jude.Uzonna@umanitoba.ca; thomas.murooka@umanitoba.ca

burden and disease reactivation[7]. These studies indicate that a balance between effector and regulatory T cell responses is established after infection and is maintained long after the skin heals, and that tipping the balance in either direction can lead to parasite clearance or disease reactivation. What remains unexplored is whether Treg suppression of effector T cell response within the chronic lesion is driven by antigen recognition, and the dynamic interplay between Tregs and effector T cells that facilitate parasite persistence. Since persistent parasites continue to undergo replication in healed skin[15], the availability of *Leishmania* antigen in healed skin to modulate helper T cell pools[18] and/or regulatory T cell functions is unclear, especially in the context of *L. major* persistence.

We have previously described that upon cutaneous *Leishmania major* infection, early Th1 responses were directed towards an immunodominant peptide (PEPCK$_{335-351}$) derived from phosphoenolpyruvate carboxykinase (PEPCK) which is conserved in all pathogenic *Leishmania* species[19]. PEPCK is expressed throughout the lifecycle of *Leishmania* and is found in the glycosomes of both promastigotes and amastigotes and elicits a dominant CD4$^+$ T cell response in both infected mice and humans[19]. We generated PEPCK-specific TCR transgenic mice on the C57Bl/6 background, where CD4$^+$ T cells recognize the PEPCK peptide$_{335-351}$ presented in the context of I-A$^b$, termed PEG mice[20]. Using two-photon microscopy, we observed robust PEG Th1:macrophage interactions during *L. major* infection in both 3D collagen and in mouse skin, validating the use of PEG T cells as a sensitive tool to examine real-time effector Th1 responses at both acute and healed stages of disease. The presence of PEG Tregs facilitated studies into how antigen specificity modulated suppressive Tregs in the context of *L. major* in vivo. Here, we show that *Leishmania*-specific Tregs displayed significantly higher suppressive activity, and that this was mediated predominantly through IL-10 production. *Leishmania*-specific Tregs did not disrupt effector Th1:macrophage conjugates during infection or alter cell surface expression involved in immunological synapse formation within infected macrophages. Finally, in vivo expansion of endogenous *Leishmania*-specific Tregs led to reactivation of disease at the primary infection site which correlated with reduced effector Th1 responses in an IL-10 dependent manner. Collectively, our data argue that the accumulation of *Leishmania*-specific Tregs in healed skin disable effector responses to promote long-term parasite persistence.

## Results

### PEPCK-specific Th1 responses against *Leishmania major* promote IFNγ production and reduce parasite load in macrophages
To measure the specificity and magnitude of PEG (PEPCK TCR transgenic) Th1 responses during *Leishmania major* infections in vivo, we first isolated naïve CD4$^+$ T cells from the spleens of both wildtype and PEG mice and activated for 48 h using anti-CD3/CD28 beads in a Th1 polarizing condition consisting of murine recombinant IL-12 and anti-IL-4 antibody. After 7 days, Th1 cells were co-cultured at 1:1 ratio with bone marrow-derived macrophages (BMM) that were either left uninfected or infected with *Leishmania major* for 6 h prior to co-culture with T cells for another 24 h. Flow cytometry analysis confirmed high IFNγ production by PEG CD4$^+$ Th1 cells, but not in control Th1 cells, whereas maximal IFNγ production was similar after PMA/ionomycin stimulation (Fig. 1A, B). Robust IFNγ responses in PEG Th1 cells were also observed in PEPCK$_{335-351}$ peptide-pulsed macrophages (Supplementary Fig. 1B). Dose response studies showed strong IFNγ production at lower infection rates (0.25:1), although maximal responses were observed at an infection rate of 10:1 (Supplementary Fig. 1A, C). ELISA analysis of culture supernatant corroborated high production of IFNγ, along with TNF, IL-2 and IL-6 (Fig. 1D). IFNγ production by effector Th1 cells promote ROS and nitric oxide (NO) production to facilitate intracellular parasite killing in macrophages[21]. BMMs were infected with GFP-expressing *L. major* for 6 h prior to co-culture with either

wildtype or PEG Th1 cells for 24 h, after which cells were stained for macrophage marker F4/80 and Hoechst nuclear dye (Fig. 1E). PEG Th1 cells significantly reduced both the percentage of infected macrophages and the average number of parasites per macrophage compared to wildtype control Th1 cells, confirming their ability to promote parasite clearance (Fig. 1F, G).

To counteract effector T cell responses, *L. major* employ mechanisms that interfere with the antigen presentation machinery[22–27]. We analyzed whether cell surface MHC-II and co-stimulatory molecule expression was altered after *L. major* infection, and whether their expression dynamics impacted effector Th1 responses in our co-culture model. We observed no reduction in MHC-II, CD80/86 and CD40 expression after infection, and similar levels of IFNγ production by Th1 cells were observed in time course studies (Supplementary Fig. 1D–H). These studies indicated that previously described dampening of antigen presentation had no measurable impact on effector T cell responses, and that PEG Th1 cells served as a sensitive tool to measure *Leishmania*-specific effector responses in vivo.

### Antigen recognition at the site of *L. major* infection leads to selective Th1 cell arrest
Intradermal infection of C57Bl/6 mice with *L. major* induces robust T cell responses that peaks at 4-6 weeks post-infection, followed by skin healing by 10–15 weeks where residual parasites remain at the primary infection site[9,28]. Intravital 2P-microscopy of the ear dermis in albino C57Bl/6 mice infected with $1 \times 10^6$ stationary phase GFP-expressing *L. major* promastigotes[29] confirmed that a localized region of infected cells can be observed for up to 20 weeks post-infection, long after the skin has healed (Supplementary Fig. 2A). To visually characterize how effector T cell responses differed during the two phases of *L. major* infection in vivo, 1-1.5 × 10$^7$ day 7 in vitro expanded control and PEG Th1 cells were stained with cell tracker dyes (CMTMR or CMAC, colors switched throughout experiments to account for possible fluorophore detection bias of our filter cubes), and adoptively transferred into each of GFP$^+$ *L. major* infected mice at weeks 3–5, or 10–15 post-infection, representing acute and healed stages of infection, respectively. Equal transfer of both populations was confirmed in the spleen of recipient mice by flow cytometry (Supplementary Fig. 2B). Intravital 2P-microscopy was performed at 1-2 days post-T cell transfer (Fig. 2A, B and Supplementary Movie 1), where both control and PEG T cells were found in infected skin at all stages, corroborating previous studies showing that Th1 cells enter the lesion environment irrespective of their antigen specificity[30]. During acute *L. major* infection, transferred control Th1 cells migrated at a mean 3D track velocity of 6.5 µm/min, whereas the mean track velocity of PEG Th1 cells was 5.1 µm/min and displayed higher cell arrest, indicative of antigen-mediated cell-cell contact formation (Fig. 2C, D). Interestingly, adoptively transferred PEG T cells also displayed lower migration speeds compared to control T cells around residual lesions observed in healed skin, suggesting that sufficient cognate antigen recognition facilitated PEG T cell deceleration. Notably, both T cell populations exhibited lower migration speeds and recruitment around persistent lesions in healed skin compared to acute infection, suggesting significant alterations in tissue architecture and immune landscape between the two infection stages (Supplementary Fig. 2C).

To further define how the two phases of *L. major* infection impacted effector Th1 migratory behaviors, mean 3D track velocity was plotted as a function of confinement ratio (track displacement/track length)[31] (Fig. 2E). All cell tracks were divided into four quadrants by setting crosshairs at a mean track speed of 5 µm/min confinement ratio of 0.4, where T cells displaying a "confined, slow" migratory behavior were defined in quadrant 1 (green shade) and T cells displaying "meandering, fast" migratory behaviors were identified in quadrant 2 (gray shade). Comparison of the relative percentages of

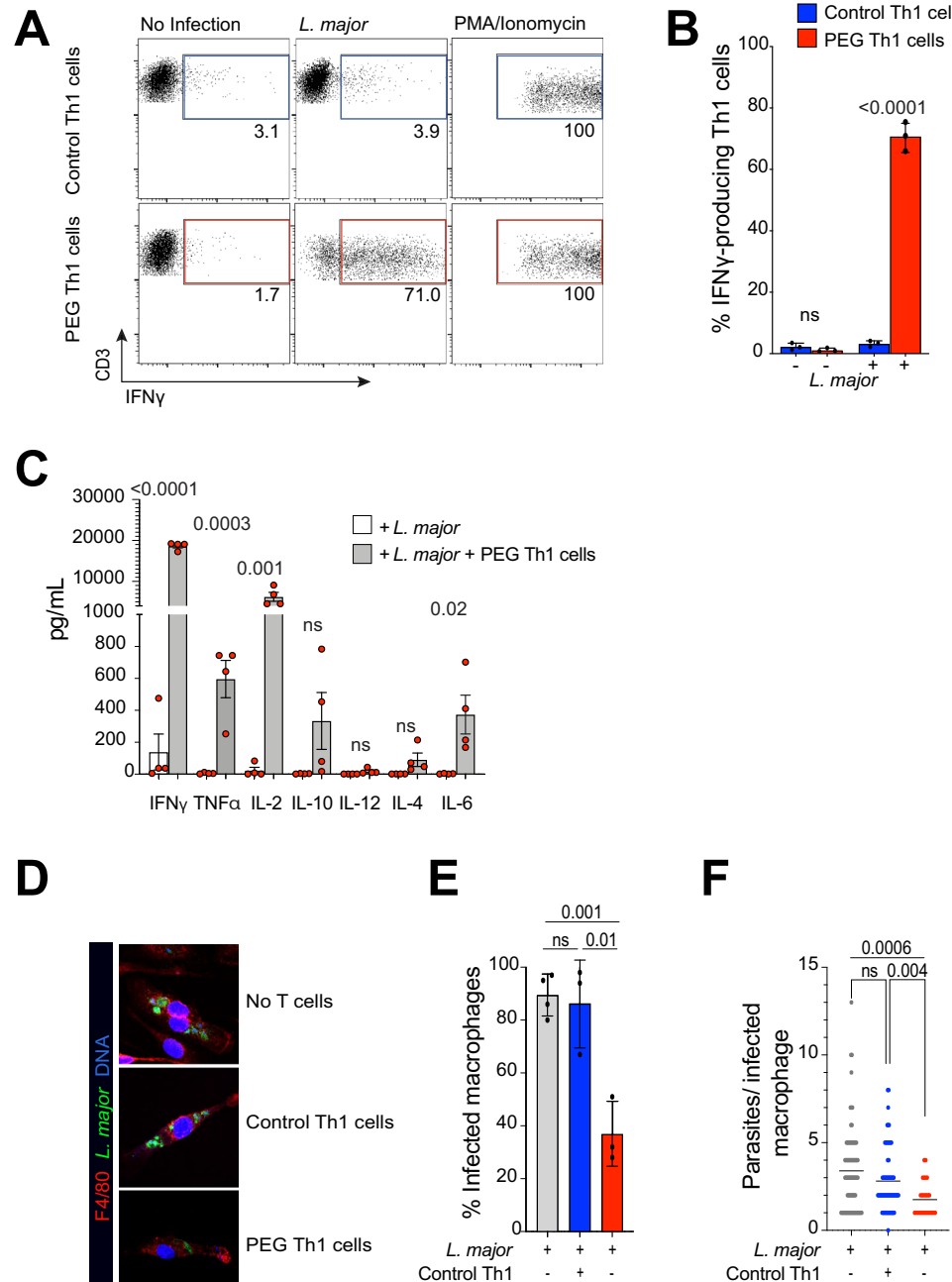

**Fig. 1 | PEG Th1 responses against *Leishmania major* are characterized by high IFNγ production and reduction of parasite load in infected macrophages.**
**A**, **B** Naïve CD4+ T cells were isolated from spleens of wildtype control or PEG C57Bl/6 mice, activated with anti-CD3/CD28 beads and expanded for 7 days in the presence of Th1 polarizing cytokines. IFNγ production was measured by flow cytometry upon co-culture with uninfected or *L. major*-infected macrophages or after stimulation of PMA/Ionomycin. Representative graph of four independent experiments is shown. Red and blue boxes represent positive IFNγ signal. Mean +/− SD. Two-tailed unpaired Student's *t* test, ns not significant. **C** Relative cytokine expression of *L. major*-infected macrophage culture alone and after PEG Th1 co-

culture was measured using multiplex ELISA. Each dot represents a mean value from 3 technical replicates, *n* = 4 independent experiments. Mean +/− SEM. Two-tailed unpaired Student's *t* test, ns not significant. **D** Representative micrographs of *L. major*-infected macrophages in the presence or absence of Th1 cells. Blue – DNA, Green – *L. major*, Red – F4/80. **E** Percent infected macrophages and **F** parasite number per infected macrophage are shown. Percent infected macrophages data constitutes of three independent experiments, parasite number per macrophage is a representative graph of three independent experiments. Two-tailed unpaired Student's *t* test, ns not significant. Source data are provided as a Source Data file.

confined slow vs meandering fast tracks showed that only 35.2% of control cells, compared to 56.2% of PEG Th1 cells, exhibited constrained behaviors indicative of cognate antigen recognition (Fig. 2F). Similarly, PEG cells displayed behaviors consistent with antigen recognition within healed skin. These data suggest that additional structural and immunological constraints modulated effector Th1 responses in healed skin.

**Regulatory T cell suppression of *anti-Leishmania* Th1 responses is enhanced through cognate antigen recognition**
Persistent *L. major* parasites limit replication to low levels in skin macrophages to evade recognition by effector T cells, thereby ensuring their long-term survival. Alternatively, low antigen availability may favor the generation and maintenance of *Leishmania*-specific regulatory T cell (Treg) in healed skin to establish an immunosuppressive

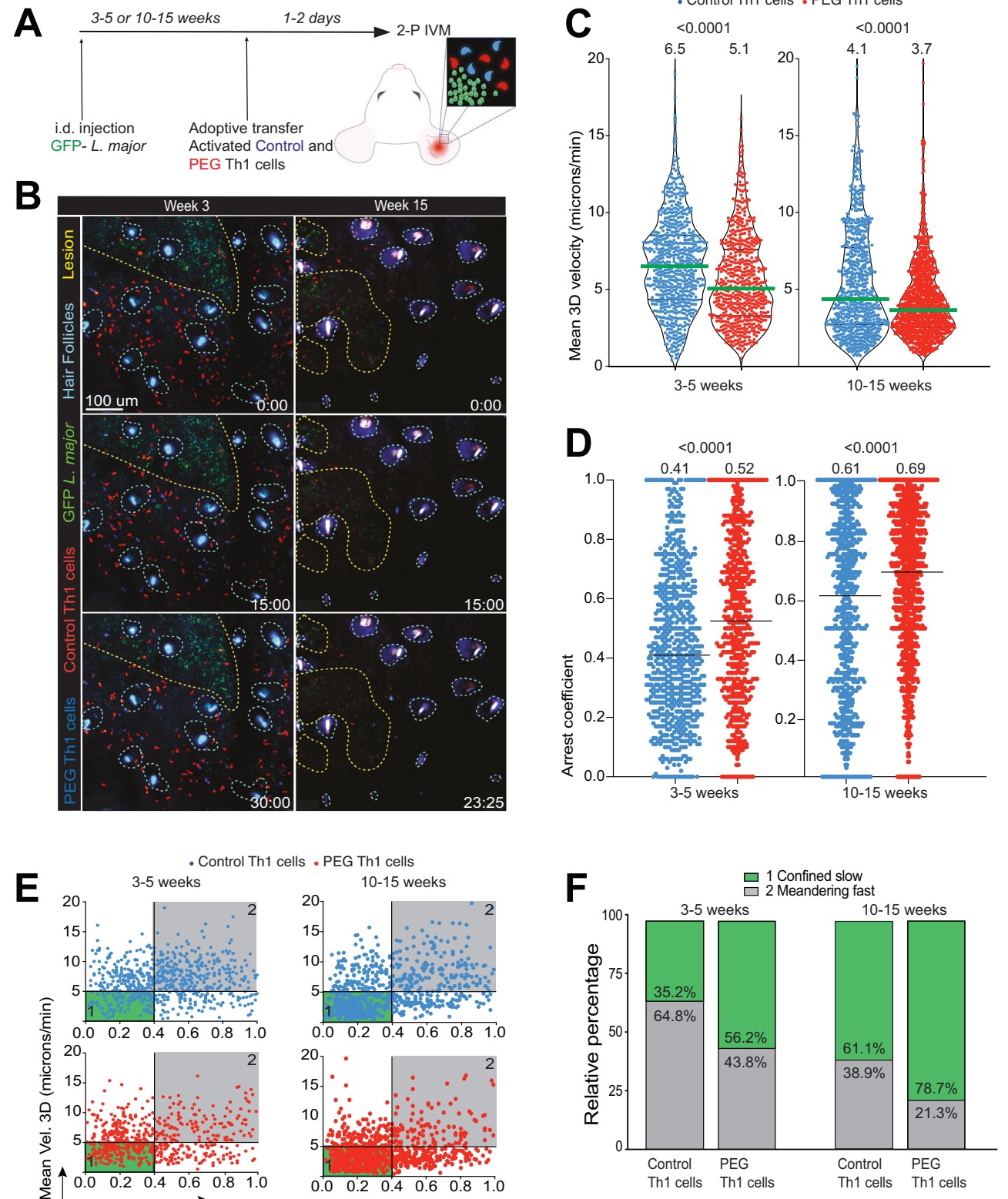

environment that favor parasite persistence. Previous studies have noted that high intralesional Foxp3 expression was associated with treatment unresponsiveness and that *Leishmania*-specific regulatory T cell (Treg)-derived cytokines were associated with active disease[9,32–34]. Thus, we hypothesize that the accumulation of Tregs in healed skin restricts effector T cell responses, allowing for low level infection to persist. We first addressed the contribution of antigen recognition on

suppressive Treg function by isolating CD4+CD25+ Tregs from the spleens of both wildtype and PEG mice for expansion using anti-CD3/CD28 beads and 2000 U/mL IL-2, as described[35]. After 7 days, phenotypic analysis of expanded Tregs from control and PEG mice showed no differences in cell number, Foxp3 or CTLA-4 expression between the two populations (Fig. 3A). PEPCK-specificity was confirmed using P3 (I-A^b-PEPCK_{335-351} tetramer) (Fig. 3A). Next, control or PEG Tregs

**Fig. 2 | In vivo cellular dynamics of Th1 response against *Leishmania major* infection. A** Experimental design of MP-IVM studies. Albino C57Bl/6 mice were infected via intradermal ear injections with 1 million GFP⁺ *L. major* parasites. At various times post-infection, 10–15 million each of control and PEG Th1 cells were stained with cell tracker dyes (CMAC blue, and CMTMR; red, alternating between experiments) and adoptively transferred into each of the recipient mice infected for the indicated time periods. Ears were prepared for intravital microscopy 24–48 h after T cell transfer. **B** Representative micrographs of the lesions after adoptive transfer, yellow and blue dotted lines represent the lesion and hair follicles, respectively. Time stamps in min:sec, scale bar: 100 μm. Mean 3D track velocity (**C**) and arrest coefficient (**D**) of the indicated Th1 cells. Blue: control cells, Red: PEG Th1 cells. Each dot represents a single cell track. Green and black lines represent median values. Data combined from 13 biologically independent mice from 8 independent experiments. Mann–Whitney *U*-test was used. **E** Track velocities of Th1 cells plotted against confinement ratio. Blue: control cells, Red: PEG Th1 cells. Each dot represents a single cell track. **F** Relative percentage of confined slow and meandering fast cell tracks. Green: confined slow, Gray: Meandering fast. Source data are provided as a Source Data file.

were co-cultured together with effector Th1 cells and *L. major*-infected BMMs to measure their suppressive activity at varying effector/Treg ratios (Fig. 3B). PEG Th1 cells were pre-stained with CMAC in order to distinguish them from Tregs by flow cytometry (Supplementary Fig. 3A). *L. major*-specific Tregs displayed significantly higher suppressive activity compared to control Tregs during infection, and this differential activity was also observed in peptide-pulsed BMMs (Fig. 3C–E, Supplementary Fig. 3B, C). Cytokine analysis of culture supernatant revealed significant elevation of IL-10 expression in the presence of PEG Tregs, whereas downregulation of pro-inflammatory cytokines was noted, including TNF, IL-1β, IL-4, IL-6, and MCP-1 (Fig. 3F, Supplementary Fig. 4). Depletion of IL-2 levels in media was also observed in the presence of *Leishmania*-specific Tregs, indicating higher IL-2 usage by these effector Tregs. The addition of anti-IL-10 monoclonal antibody to the Teff/Treg co-cultures fully restored IFNγ production by effector T cells, implicating IL-10 as the major driver of Treg suppressive activity in our ex vivo model (Fig. 3G). Notably, no significant changes in TGFβ production were observed (Supplementary Fig. 4). Addition of rmIL-10 alone was sufficient to inhibit effector Th1 cells IFNγ expression in a dose-dependent manner, whereas the addition of TGF-β had a minimal effect (Supplementary Fig. 3D). These results provide direct evidence that *Leishmania*-specific regulatory T cells display enhanced suppressive function during parasitic infection in an IL-10 dependent manner.

As IL-10 production significantly suppressed IFNγ secretion by effector PEG Th1 cells, we examined whether Tregs reduce intracellular parasite killing efficiency in macrophages. BMMs were infected with GFP-expressing *L. major* for 6 h prior to co-culture alone or with PEG Th1 alone, PEG Th1⁺ control Tregs, or PEG Th1⁺ PEG Tregs for 24 h, after which cells were stained for macrophage marker F4/80, GFP, and Hoechst nuclear dye (Fig. 3I). Reduction of both the percentage of infected macrophages and the average number of parasites per macrophage by the PEG Th1 cells was significantly restored by PEG Tregs and to a lesser degree control Tregs, confirming their ability to restrain Th1 mediated parasite control (Fig. 3J, K).

### Suppression of anti-*Leishmania* T cell responses by Tregs does not involve disruption of Teff:BMM contacts

To isolate cell-intrinsic effector T cell behaviors from environmental and cellular constraints imposed by the infectious skin niche, we first used a reductionist approach to visualize T cell:BMM interaction dynamics and effector responses using a 3D fibrillar collagen matrix model[36]. Celltracker blue-labeled BMMs were either left uninfected or infected with GFP⁺ *L. major* for 24–72 h prior to co-embedding with PEG Th1 cells (red) and control Th1 cells (green) in 3D collagen and imaged at 30-minute intervals, as we have done previously[36] (Fig. 4A). In the absence of infection, Th1:BMM contacts were short-lived, with similar contact times observed regardless of antigen specificity (Fig. 4B, C and Supplementary Movie 2). As expected, PEG Th1 cells engaged in stable contacts with *L. major* infected macrophages (Fig. 4B, C), correlating with a decrease in PEG Th1 cells mean 3D velocity (Fig. 4D). Next, we evaluated how *L. major*-specific Tregs impacted Th1:BMM interaction dynamics by co-embedding infected BMM (red), PEG Th1 (blue) and Celltracker green labeled Tregs (either control or PEG Tregs) in 3D collagen (Fig. 4A, E). As predicted, PEG

Tregs also formed prolonged contacts with infected BMMs leading to a reduction in their overall mean 3D track velocity (Fig. 4F, G). Surprisingly, the presence of *Leishmania*-specific Tregs did not disrupt effector T cell contacts with infected macrophages (Fig. 4H, pink box) but maintained prolonged contacts, often resulted in the formation of large cell clusters consisting of all three cell types (Fig. 4E lower panels, Supplementary Movie 3). Notably, we observed an overall reduction in effector Th1 cell migration speeds in the presence of infection, which was consistent with in vivo behaviors during the healed stages of *L. major* infection. The presence of infected macrophages led to a higher proportion of Th1 cells displayed confined behavior (Fig. 4J, K), whereas Tregs increased the proportion of arrested cells equally, regardless of their specificity (Fig. 4L, M).

The immunological synapse creates a stable interface between T cells and APCs during cognate antigen stimulation and involves formation of TCR:MHC-II and CD28:CD80/86 receptor complexes[37,38]. We postulated that Treg:BMM contacts may augment cell surface receptor expression, indirectly facilitating enhanced effector:BMM interactions. To test this, infected BMMs were co-cultured with effector PEG Th1 cells in the presence or absence of Tregs and analyzed for MHC-I, MHC-II, CD40, CD80, and CD86 expression (Fig. 4N). We observed no differences in their expression on macrophages in the presence of either Treg populations, ruling out modulation of antigen presentation as a mechanism to induce large BMM:T cell clusters during *Leishmania major* infection.

### In vivo expansion of *L. major*-specific Tregs leads to disease reactivation

In vivo Treg depletion leads to sterilizing immunity against *L. major*[9], whereas we have previously shown that Treg expansion leads to reactivation of disease in healed mice[13]. We used the latter approach to mechanistically address how Tregs modulated effector T cell responses to support long-term parasite survival and to address the role of cognate antigen in establishing a suppressive environment in healed skin. To this end, Foxp3-GFP mice were intradermally infected in the ear pinna with 10⁶ dsRed⁺ *L. major* promastigotes[39] and allowed to heal. At day 60 post-infection, mice were challenged with either PBS, 5 × 10⁶ heat-killed wildtype *L. major* promastigotes or 5 × 10⁶ heat-killed PEPCK-deficient *L. major* promastigotes[20] via footpad injection[13], and intravital microscopy of the previously healed ear dermis was performed after 2 weeks (Fig. 5A). Expansion of Foxp3-GFP⁺ cells was observed in mice challenged with heat-killed parasites, with many found robustly migrating within the primary infection site (visualized by the lack of blue collagen structures; Fig. 5A, B and Supplementary Movie 4). High lesion score[40] and significant redness/swelling was observed after challenge with heat-killed wildtype *L. major* but observed to a lesser extent after challenge with PEPCK⁻/⁻ *L. major* (Fig. 5C, D), consistent with higher parasite burden in the former scenario (Fig. 5E). While no differences in CTLA-4, PD-1, and CD44 expression was observed in Tregs present in the lesion site across the different groups (Fig. 5F), there was a lack of PEPCK-specific Treg expansion in mice challenged with PEPCK-deficient parasite (Fig. 5G), indicating that optimal suppressive Treg function required PEPCK recognition in healed skin. Indeed, the lack of PEPCK-specific Tregs in healed skin failed to suppress IFNγ production by endogenous CD4⁺

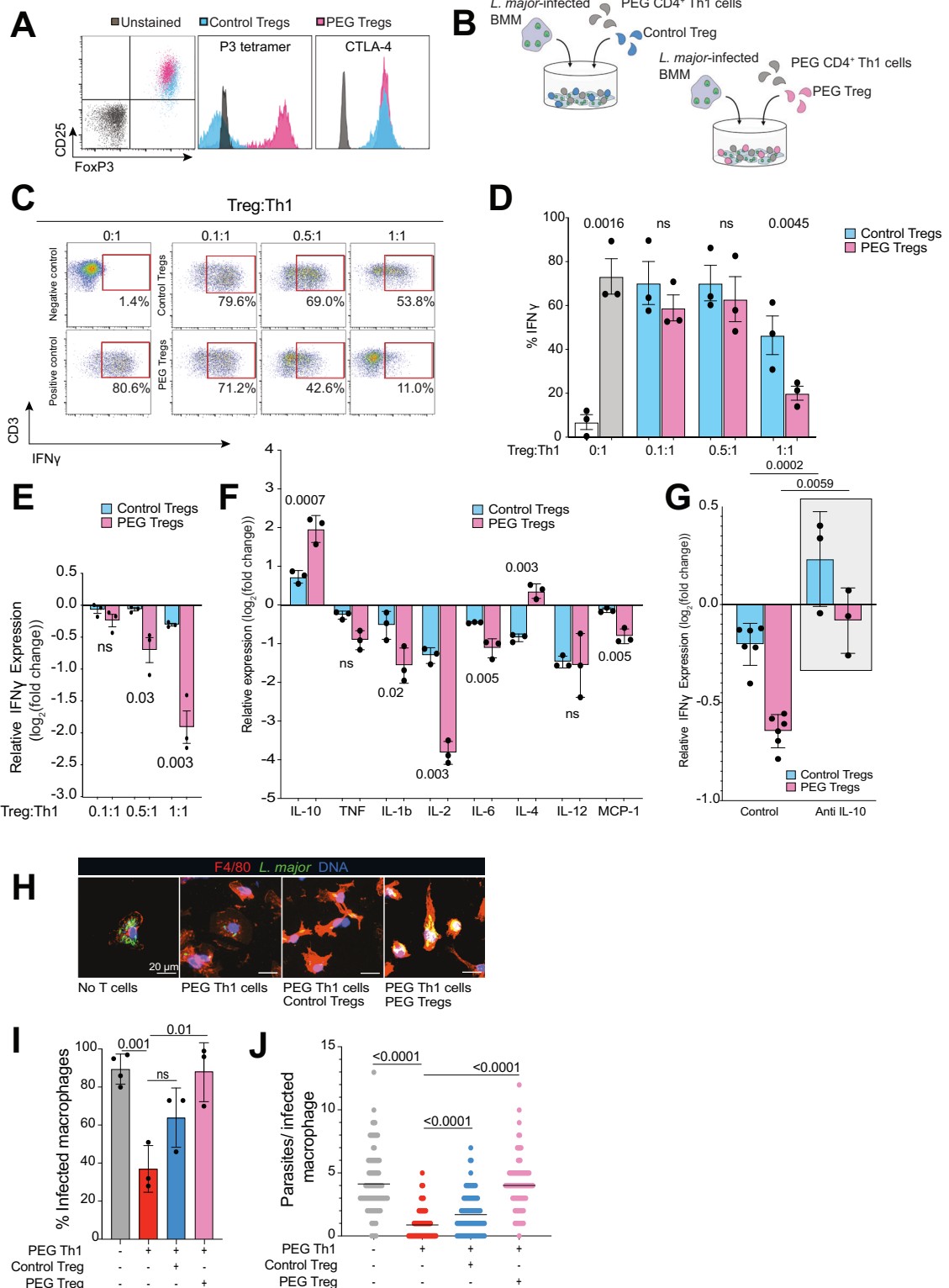

Tbet[+] effector T cells compared to those expanded with wildtype parasites (Fig. 5H). These data provide direct evidence that Tregs actively suppress effector T cell responses in healed skin, which is enhanced by *L. major* antigenic stimulation. Given that our co-culture studies indicated that IL-10 mediated strong suppressive function by *L. major*-specific Tregs in vitro, we next addressed whether similar mechanisms were at play in vivo. Healed mice were challenged with either PBS or heat-killed *L. major* promastigotes as before, but also in the presence of anti-IL-10 monoclonal antibody given intraperitoneally

2 days prior (Fig. 6A). IL-10 blockade completely restored robust IFNγ production by effector CD4[+] T cells in skin (Fig. 6B, C). We confirmed that IL-10 was produced by expanded Tregs at the lesion site and not by effector T cells in the ear pinna (Fig. 6D). In order to evaluate whether IL-10 blockade also resulted in a lower parasite burden, mice that have healed from a prior footpad *L. major* infection were challenged with either PBS or heat-killed *L. major* promastigotes into the contralateral footpad, with some animals given anti-IL-10 monoclonal antibody intraperitoneally every 2 days over 2 weeks. (Fig. 6E, F).

**Fig. 3 | Antigen-specific Tregs suppress Th1 response against *Leishmania major* infection. A** Regulatory T cells were isolated from spleens of wildtype control or PEG C57Bl/6 mice activated with anti-CD3/CD28 beads and expanded for 7 days in the presence of IL-2. **B** Experimental design of the Treg suppression assay. PEG Th1 cells were co-cultured with *L. major*-infected macrophages and either control or PEG Tregs for flow cytometry and multiplex cytokine analysis. **C**–**E** IFNγ production by PEG Th1 cells was measured by flow cytometry at various Th1: Treg ratios. **C** Red box: positive IFNγ signal. **D** % IFNγ production and **E** relative IFNγ expression compared to % IFNγ production by PEG Th1 cells without Tregs (gray bar). White bar: PEG Th1 cells cultured with uninfected macrophages (negative control), gray bar: PEG Th1 cells cultured with *L. major*-infected macrophages. Blue and pink bars: Control and PEG Tregs, respectively, added to culture with PEG Th1 cells and *L. major*-infected macrophages. Data shown are combined from three independent experiments; each dot represents a mean value from three technical replicates. Mean +/− SEM. Two-tailed unpaired Student's *t* test, ns not significant. **F** Relative expression of cytokines measured by multiplex ELISA during Treg suppression assays. Control (blue) and PEG (pink) Tregs. Data combined from three independent experiments; each dot represents a mean value from three technical replicates. Mean +/− SEM. Two-tailed unpaired Student's *t* test, ns not significant. **G** IFNγ production by PEG Th1 cells in the presence of Control (blue) and PEG (pink) Tregs at 1:1 Th1:infected macrophage ratios. Gray bar: anti-IL-10 antibody was added to the co-culture. Each dot represents a single technical replicate, data representative of 2 independent experiments. Two-tailed unpaired Student's *t* test. **H** Representative micrographs of *L. major*-infected macrophages alone, or cultured with PEG Th1 cells, PEG Th1 cells and control Tregs, or PEG Th1 cells and PEG Tregs. Blue – DNA, Green – *L. major*, Red – F4/80. **I** Percent infected macrophages and **J** parasite number per infected macrophage are shown. Percent infected macrophages data constitutes of two independent experiments, parasite number per macrophage is a representative graph of two independent experiments. Two-tailed unpaired Student's *t* test, ns not significant. Source data are provided as a Source Data file.

Consistent with observations in the ears, challenge with heat-killed *L. major* induced disease reactivation that was accompanied by a significant increase in tissue parasite burden. Importantly, in vivo IL-10 blockade resulted in lower parasite load, again implicating this cytokine as a major mechanism of effector Treg suppressive function during parasite persistence.

## Discussion

Despite the generation of strong Th1 responses after cutaneous *Leishmania major* infection, parasites in healed skin establish an equilibrium with the surrounding environment to maintain a constant pool of persistently infected cells. An important question is whether chronically infected cells continue to present *L. major* antigens, and if so, whether ongoing Th1 responses are counterbalanced by the presence of suppressive regulatory T cells to ensure survival of residual parasites in healed skin. Here, we used novel *Leishmania*-specific TCR transgenic mice as a sensitive tool to directly evaluate both antigen-specific effector and immunosuppressive T cell responses in healed skin using two-photon microscopy. We show that *Leishmania*-specific Tregs displayed significantly higher suppressive activity compared to polyclonal control Tregs, and that this was mediated through IL-10 production and not through disrupting cell-cell contacts or modulation of the antigen presentation machinery. In vivo expansion of *Leishmania*-specific Tregs within healed lesions was followed by disease reactivation that was also dependent on IL-10 release. Interestingly, the lack of Treg expansion that recognized the immunodominant peptide PEPCK was sufficient to restore robust effector Th1 responses and better parasite control. Together, our data suggest a stochastic model of *Leishmania major* persistence in skin, where cellular and molecular factors that control parasite numbers are modulated by the presence of *Leishmania*-specific Tregs that need to be overcome to achieve complete parasite clearance.

This study visually characterized effector T cell migratory responses during the acute and healed stages of *Leishmania major* infection using two-photon microscopy. Prior microscopy studies utilized CD4 T cells specific for the *Leishmania* homolog of receptors for activated C kinase (LACK) epitope in Balb/c mice, where the local immune response is insufficient to control the infection[30]. The strength of the present study is the ability to track PEPCK-specific T cells in healing C57Bl/6 mice, which unlike LACK, is expressed by parasites during their entire lifecycle and is targeted by CD4 T cell responses in humans[19]. We observed that PEG Th1 cells readily homed to infectious lesion sites and engaged with *L. major*+ cells, leading to arrested behaviors compared to non-specific Th1 cells. When transferred into mice with healed lesions, their homing ability was much less efficient, and both T cell populations displayed significantly reduced motility profiles at the lesion site suggesting significant structural remodeling at this tissue site. At steady-state, most peripheral tissues are characterized by a dense extracellular matrix (ECM) that modulates T cell search behaviors based on their density and composition[41]. During an inflammatory response, increased protease secretion and resulting ECM turnover can loosen these stromal networks to facilitate αV integrin-dependent T cell migration, specifically $α_vβ_1$ and/or $α_vβ_3$[42]. Robust αV integrin expression by effector Th1 cells has been reported during acute *L. major* infection[42] and may also guide migration of Th1 cells during peak infection irrespective of antigen specificity, as previously reported[30]. To our knowledge, only one study has looked at the ECM composition at the chronic stages of infection[43]. During the tissue remodeling stage after removal of invading pathogens, newly synthesized collagen type III is gradually replaced with the dominant collagen type I. However, healed skin after *L. amazonensis* infection comprises mostly of collagen type III, which deposits around the parasitized macrophages and these deposits are even more prominent in susceptible BALB/c mice[43]. How these structural changes alter T cell trafficking and migratory behaviors during *Leishmania* infection is unclear, but the tumor microenvironment is often enveloped by remodeled ECM that restrict T cells from entering the tumor mass[44,45]. The extent of ECM remodeling in healed skin by factors such as IL-10 or TGF-β and how they impact T cell access and effector responses remains unclear and subject to our ongoing investigations[46].

Aside from structural remodeling that can impact effector T cell responses in skin, *L. major* infection results in significant immunological changes that can further blunt T cell responses. Adoptive transfer of naïve PEG cells into mice prior to infection does not lead to sterilizing immunity, indicating that generation of strong effector Th1 immunity is insufficient to achieve complete parasite clearance. Early studies demonstrated that Treg depletion leads to sterilizing immunity, underscoring their role in this process[9]. Tregs found in healed *L. major* lesions were largely parasite-specific, where those recovered from healed C57Bl/6 mice produced IL-10 upon restimulation with *Leishmania* antigens. Treg adoptive transfer into wild-type or RAG$^{-/-}$ mice exacerbated lesion development and parasite growth, whereas co-transfer of Tregs with effector T cells into RAG$^{-/-}$ mice lead to a reduction in IFNγ+ T cell numbers and higher parasite numbers, mirroring the natural course of infection in C57Bl/6 mice[9,47]. Importantly, Treg-mediated suppression was not dependent on tissue source (isolated from healed skin vs lymph node of naïve mice) and were equally expanded and suppressive following *L. major* infection[17]. Regulatory T cells from human lesions displayed higher CCR5 expression, which corresponded with observations that CCR5-deficient mice can completely clear *Leishmania major* infection[48,49]. In a model where galectin-3 deficiency increases the frequency of peripheral Tregs, the severity and parasite burden following *L. major* infection was increased significantly[50]. Finally, ablation of Langerhans cells decreased Treg numbers and IL-10 levels which was associated with enhanced Th1 responses[51], whereas inactivation of the aryl hydrocarbon receptor

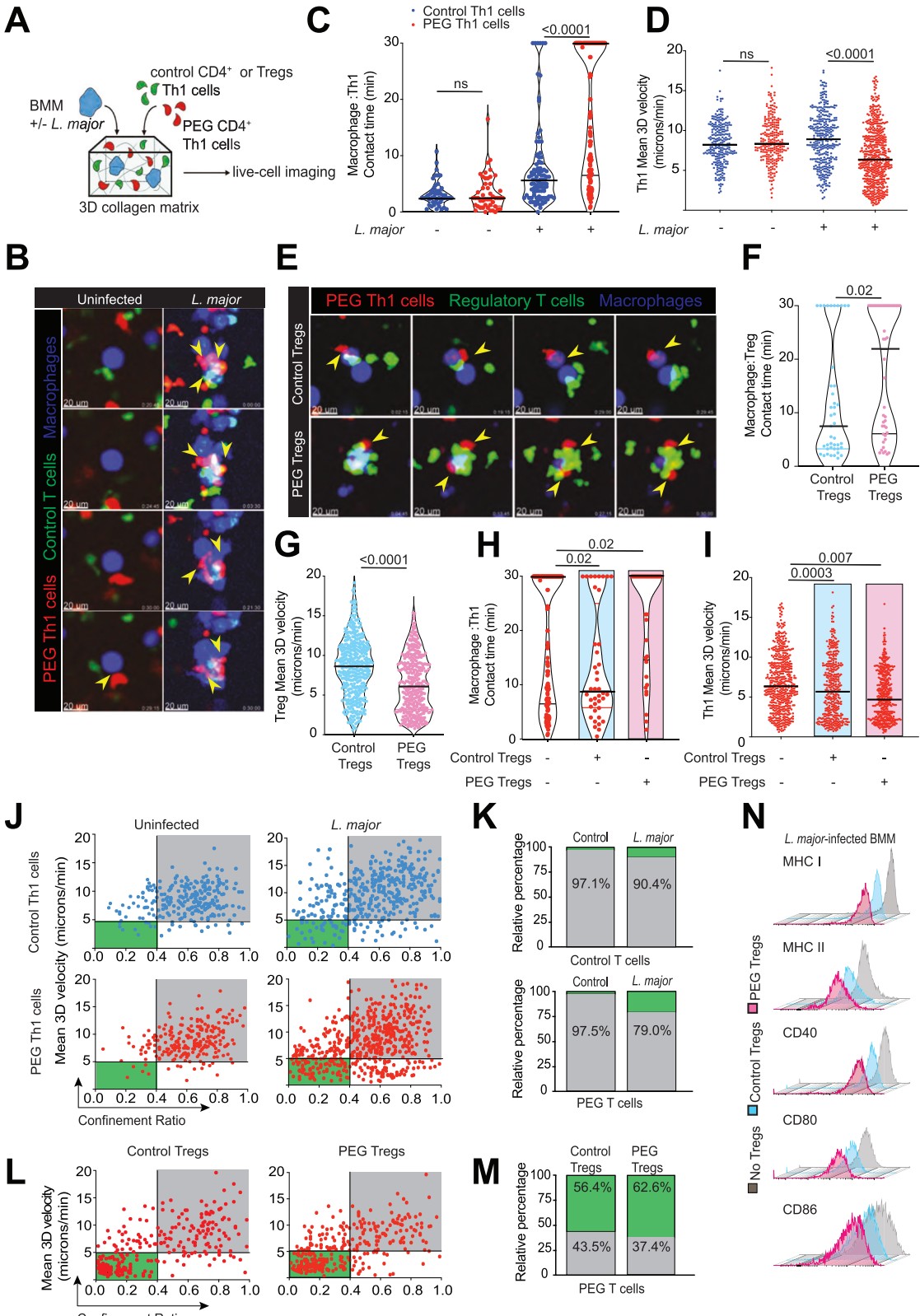

results in a reduction in Tregs, resulting in higher IFNγ production and improve resolution of *L. major*[52]. Our data further extends these studies by directly showing that Tregs specific for the immunodominant *L. major* peptide PEPCK display strong suppressive activity in vitro and in vivo, and that this was through IL-10 production. We provide compelling data that recruitment of PEPCK-specific Tregs are major players in establishing an immunosuppressive environment in healed lesions

by blunting effector Th1 responses. Tregs in healed *Leishmania* lesions have been speculated to come from the draining lymph node[9,49], which is supported by our data where large PEPCK-specific Treg numbers are found in infected skin compared to cervical lymph nodes.

Tregs exhibit numerous mechanisms of suppression, ranging from limiting T cell expansion in the lymph node to attenuating effector responses at peripheral tissue sites. Tregs can reduce the

**Fig. 4 | PEG Regulatory T cells do not disrupt prolonged macrophage:PEG Th1 contacts. A** Control Th1 cells or Tregs and PEG Th1 cells were stained with cell tracker dyes (CMFDA; green, and CMTMR; red, respectively) and co-cultured with stained uninfected or *L. major*-infected macrophages (CMAC; blue). **B** A time series micrograph of co-cultures of macrophages, control and PEG Th1 cells in the absence or presence of *L. major* infection at indicated times. Time stamps in min:sec represent elapsed time of the recordings. **C** Real-time contact duration between control (blue dots) or PEG (red dots) Th1 cells and macrophages in the absence or presence of *L. major* infection. Each dot represents a single contact. Black lines represent median values. Statistical analysis: Mann–Whitney *U*-test, ns not significant. Data combined from three independent experiments. **D** Mean 3D track velocity of control (blue dots) or PEG (red dots) Th1 cells cultured with macrophages in the absence or presence of *L. major* infection. Each dot represents a single cell track. Black lines represent median values. Statistical analysis: Mann–Whitney *U*-test, ns not significant. Data combined from three independent experiments. **E** A time series micrograph of co-cultures of *L. major*-infected macrophages, PEG Th1 cells in the presence of control or PEG Tregs at indicated times. Time stamps in min:sec represent elapsed time of the recordings. Real-time contact duration between control or PEG Tregs and macrophages (**F**) and mean 3D track velocity (**G**), each dot representing a single contact. Black lines represent median values. Statistical analysis: Mann–Whitney *U*-test. Data combined from three independent experiments. Source data are provided as a Source Data file. **H** Real-time contact duration between PEG (red dots) Th1 cells and *L. major*-infected macrophages. Each dot represents a single contact. Black lines represent median values. Statistical analysis: Mann–Whitney *U*-test, ns not significant. Data combined from three independent experiments. **I** Mean 3D track velocity of PEG (red dots) Th1 cells. Each dot represents a single cell track. Black lines represent median values. Statistical analysis: Mann–Whitney *U*-test, ns not significant. Data combined from three independent experiments. Track velocities of control (blue) or PEG (red) Th1 cells cultured with uninfected or *L. major*-infected macrophages plotted against confinement ratio (**J**) and relative percentage of confined slow and meandering fast tracks (**K**). Data combined from three independent experiments. Track velocities of PEG (red) Th1 cells cultured with *L. major*-infected macrophages in the presence of control or PEG Tregs plotted against confinement ratio (**L**) and relative percentage of confined slow and meandering fast tracks (**M**). Data combined from three independent experiments. **N** Flow cytometry analysis of *L. major*-infected macrophage surface molecules. Representative data from three independent experiments. Source data are provided as a Source Data file.

frequency of stable contacts between dendritic cells and naïve T cells in the lymph node, thereby lowering T cell activation[53] or remove cognate MHC-II-peptide complexes in an antigen-specific manner through trogocytosis[54]. CD80 and CD86 co-stimulatory molecules can also be selectively removed during DC-Treg contacts, also reducing T cell activation levels[55,56]. A surprising finding was that PEG Tregs neither disrupted cell-cell interaction dynamics between effector Th1 and infected macrophages in vitro, nor did they alter antigen presentation or CD80/86 expression. Cognate antigen recognition strengthened PEG Treg suppressive function through high IL-10 production without altering effector T cell migratory behaviors. We interpret these data as Treg-derived IL-10 having a broad, suppressive bystander effect on nearby CD4+ T cells in a manner that is reminiscent of effector Th1 cells exerting protective effects through IFNγ release at a considerable distance from the site of antigen presentation[57]. IL-10-mediated regulation of CD4+ T cells likely works in concert with their ability to promote M2 characteristics in dermal macrophages that amplify regulatory functions and permits long-term parasite residence[58–60].

Tregs facilitate the transition from an inflammatory to a reparative phase during the immune response that involves secretion of IL-10 and TGFβ[61]. However, efforts to mitigate collateral damage from excessive inflammation may be co-opted by *Leishmania* to establish a pool of persistently infected cells by recruiting *L. major*-specific Tregs. This not unique to *Leishmania*, as an increasing number of studies show that pathogen-specific Tregs are detected in infectious settings. Pulmonary infection with *M. tuberculosis* (Mtb) induced an early expansion of activated Tregs in the lymph node that recognized an immunodominant Mtb epitope that blunted protective T cell recruitment in mice[62,63]. Tregs that recognize the immunodominant CD4+ T cell epitope are found in the brains of rJ2.2 strain of MHV-infected mice, where they inhibited pathogenic CD4+ T cell responses to ameliorate encephalitis severity[64]. In contrast to our observations, a study did not observe 2 W:I-A^b-specific Foxp3+ Tregs at the healed lesion site after infection with a recombinant *L. major* expressing the chimeric protein 2 W (*Lm*-2W)[18]. The major difference in approach is that we measured endogenous responses to an immunodominant, naturally processed PEPCK protein that is essential for *L. major* replication in vivo, using tetramer staining of isolated cells ex vivo. Our previous studies clearly demonstrate that Th1 responses are focused on a relatively narrow range of immunodominant epitopes, including PEPCK, that dominate anti-parasitic T cell immunity during the acute phase of disease[19]. The current study provides new evidence that PEPCK-specific immunosuppressive Treg responses are also directed towards

immunodominant epitopes, and that they equally impose a dominant suppressive role that facilitate long-term parasite persistence in healed skin. The exclusive use of male mice prevents us from making broad conclusions on T cell regulation across both sexes and is a limitation of this study. Comparative studies in both male and female recipients will permit investigations into whether biological sex is an important regulator of Treg function within healed *L. major* lesions, and is our current investigative focus.

We show that the majority of IL-10 was produced by regulatory T cells in healed lesions, and that in vivo IL-10 blockade resulted in stronger effector responses. However, other studies have shown that IL-10 can be produced by non-regulatory T cells. Studies with the non-healing strain of *L. major* NIH/Seidman (Sd) showed that Th1 cells produced IL-10 along with high IFNγ production compared to the FV1 strain[28]. IL-10 production by the CD25^- Foxp3^- subset was necessary for the evolution of a non-healing phenotype in mice, although removal of IL-10 producing Tregs also exacerbated disease[65]. Similarly, both Foxp3+ and Foxp3^- CD4+ T cells produced IL-10 upon infection in IL-4α^−/− BALB/c mice[66]. Infection with *L. major* expressing the chimeric protein 2 W (*Lm*-2W) also reported IL-10 producing 2W-specific Th1 cells[18]. While these studies highlight that effector T cell subsets can produce IL-10 depending on the disease and environmental context, we observed that Tregs, and not Th1 cells, were the main source of IL-10 during chronic *Leishmania major* infection in C57Bl/6 mice. In this scenario, high IL-10 production by a relatively limited number of parasite-specific Tregs may be sufficient to enforce an immunomodulatory setting during persistent *Leishmania major* infection. Our data suggest that selective removal of immunodominant Tregs may be sufficient to facilitate improved parasite clearance.

## Methods
### Mice
All animal studies were approved by the Animal Care Committee (ACC) at the University of Manitoba in accordance with the Canadian Council for Animal Care guidelines (protocol #21-022). Male wild type and albino C57BL/6J mice (6–8 weeks old) were obtained from the University of Manitoba Central Animal Care Services (CACS) breeding facility and Jackson Laboratory, respectively (stock 00058). FoxP3-GFP male mice were purchased from Jackson Laboratory (stock 006772 homozygous). PEPCK TCR-transgenic mice on the C57LB/6J background were acquired from an in-house breeding colony from CACS (University of Manitoba). All mice were maintained in a specific pathogen-free environment under containment level 2+ biosafety precautions.

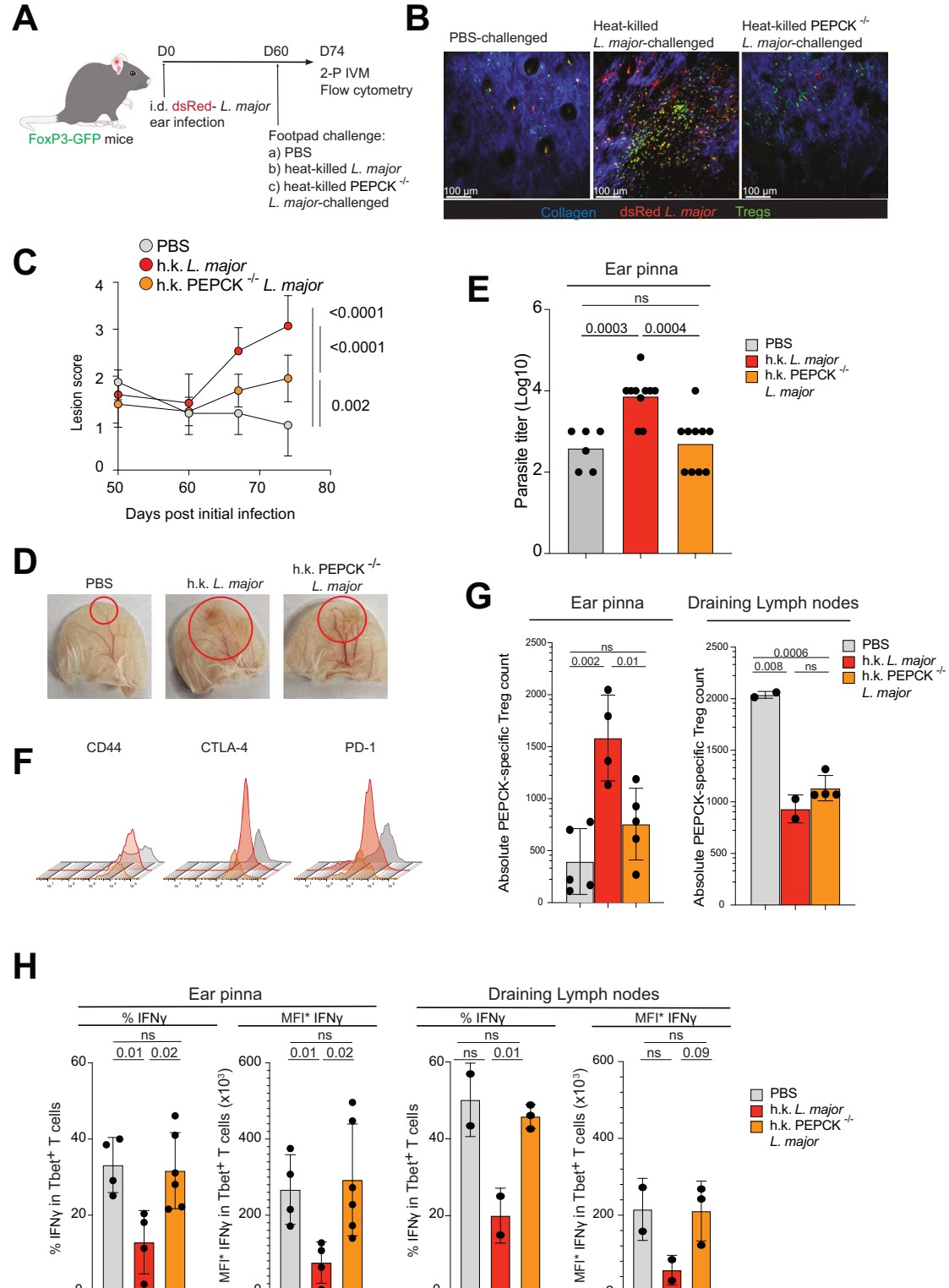

## Parasites, infections, challenge (primary and secondary), and parasite burden

Wild-type (WT), GFP, and dsRed *L. major* MHOM/80/Friedlin were grown in M199 medium supplemented with 20% heat-inactivated FBS, 100 U/ml penicillin, and 100 µg/ml streptomycin (complete M199) in a 27 °C parasite incubator. For infection, day-7 stationary-phase promastigotes were washed three times with sterile PBS and counted using an optical microscope at ×100 magnification. C57BL/6 mice were infected in the ear pinna with 10 µL of PBS containing $1 \times 10^6$ stationary phase promastigotes. In some studies, 12 or 17-wk infected C57BL/6 were challenged with $5 \times 10^6$ autoclaved[67] promastigotes in the footpads. For IL-10 blockade studies, 300 µg of anti-IL-10 antibody was injected intraperitoneally 2 days prior to sacrifice or 200 µg every 2 days over 2 weeks. Parasite burden in infected ears and footpads was determined by limiting dilution assay[68]. Briefly, ears or footpads were collected and homogenized in 2 ml complete parasite medium using 15 ml tissue grinders (VWR, Edmonton, AB, Canada). The suspension was then

**Fig. 5 | Inoculation of healed mice with killed *L. major* parasites induces Treg expansion and lesion reactivation. A** C57Bl/6 Foxp3-GFP mice were infected via intradermal ear injections with 1 million dsRed⁺ *L. major* parasites. 60 days post infection, mice were challenged with PBS or $5 \times 10^6$ heat-killed wild-type or PEPCK$^{-/-}$ *L. major* parasites into the footpad. The ears were prepared for intravital microscopy or flow cytometry analysis 14 days post challenge. **B** Representative micrographs of the lesions post challenge. Blue: collagen (SHG), red: dsRed⁺ *L. major*, Green: Foxp3-GFP cells. **C** Lesion scores of ears post challenge. Statistical analysis of the last timepoint: Two-tailed unpaired Student's *t* test. Mean +/− SD. Data combined from three independent experiments. **D** Pictures of albino C57Bl/6 ears 14 days post challenge, red circles highlighting lesion area. **E** Parasite burden from ears of healed mice challenged with PBS, or $5 \times 10^6$ heat-killed wild-type or PEPCK$^{-/-}$ *L. major* parasites into the footpad. Each dot represents data from one ear. Two-tailed unpaired Student's *t* test, ns not significant. **F** Phenotypic analysis of Tregs in lesion skin. Gray: Tregs extracted from PBS-challenged mice, Red and Yellow: Tregs extracted from mice challenged with WT or PEPCK$^{-/-}$ heat-*killed L. major*, respectively. **G** Absolute cell counts of PEPCK-specific CD45⁺CD3⁺CD4⁺Foxp3⁺ cells in the ears and draining lymph nodes. Each dot represents a single ear or draining lymph node. *n* = 2 biologically independent mice for h.k. *L. major* challenge, 3 for PBS-challenged mice, and 4 for h.k. PEPCK$^{-/-}$ *L. major* challenge. Two-tailed unpaired Student's *t* test, ns not significant. Mean +/− SD. Data combined from two independent experiments. **H** Percent and MFI* (Absolute expression, MFI × %IFNγ⁺) of IFNγ production in CD45⁺CD3⁺CD4⁺ Tbet⁺ T cells in the ears and draining lymph nodes. Each dot represents a single ear/ draining lymph node/spleen, *n* = 2 biologically independent mice for PBS or h.k. *L. major* challenge, and 3 for h.k. PEPCK$^{-/-}$ *L. major* challenge. Two-tailed unpaired Student's *t* test, ns not significant. Mean +/− SD. Source data are provided as a Source Data file.

plated in 96-well plates in triplicates at 10-fold serial dilution, incubated for 7 days at 27 °C and assessed for parasite growth under a microscope.

### In vitro generation and infection of bone marrow−derived macrophages

Bone marrow−derived macrophages (BMMs) were differentiated from stem cells obtained from the femur and tibia of a naive C57BL/6 mice. Briefly, the bones were separated from the surrounding muscles and the content of the bones was flushed with 5 ml of RPMI 1640 into a polypropylene petri dish using a syringe and a 25-G needle and made into single-cell suspensions. Following depletion of RBCs with ACK lysis buffer (150 mM NH₄Cl, 10 mM KHCO₃, 0.1 mM Na₂EDTA [pH 7.2−7.4]), cells were seeded at a density of $25 \times 10^6$/ml in T75 ml flasks in complete RPMI 1640 medium supplemented with 10% FBS (VWR Seradigm), 2 mM GlutaMAX (Gibco), 1 mM sodium pyruvate (Corning Cat #25-000-CI), 10 mM HEPES (Sigma-Aldrich) and β-Mercaptoethanol with 25ng/mL M-CSF (Biolegend Cat# 574808) for 72 h at 37 °C in a CO₂ incubator. At 72 h adhered cells reached confluency of 80-90% and non-adhered cells were washed off with PBS. BMMs were detached using Accutase® Cell Detachment Solution (Biolegend Cat# 423201) for 15 min at 37 °C in a CO₂ incubator. Detached BMMS were infected with distinct parasite strains at a cell/parasite ratio of 1:10. After a 6-h incubation (at 37 °C), the cells were washed twice (centrifuged at 600 rpm for 5 min) to remove free parasites.

### In vitro generation and polarization of Th1 cells

Spleens from wild type or PEG C57Bl/6 mice were excised and mashed with a 40 µm mesh nylon strainer, washed with PBS, and resuspended at a concentration of 1*10⁸ cells/mL in PBS supplemented with 2% FBS (VWR Seradigm cat #1500-500) and 1 mM EDTA (Sigma-Aldrich, cat #E7889). Naïve T cells were isolated using Stemcell™ EasySep™ Mouse Naïve CD4⁺ T Cell Isolation Kit (Cat# 19765) and re-suspended at a concentration of 1*10⁶ cells/mL in complete RPMI media in a 24 well plate with 25µL/mL Dynabeads® Mouse T-Activator CD3/CD28 (Thermo Fisher Scientific Cat# 11456D), 20 ng/mL recombinant human IL-12 (Biolegend Cat# 577002) and 5 µg /mL anti-mouse IL-4 antibody (Biolegend Cat# 504108). After 48 h of incubation at 37 °C, Dynabeads® were removed from T cells using Stemcell™ EasySep™ Magnet (Cat# 18000). At day 2 cells were counted and resuspend at $0.2 \times 10^6$ cells/mL in complete RPMI and 20 ng/mL IL-12, 5 µg/mL anti-mouse IL-4, and 25 ng/mL IL-2 (Peprotech Cat# 200-02). After 48 h of incubation cells were resuspended in complete RPMI supplemented with 25 ng/mL IL-2 at $0.5 \times 10^6$ cells/mL. Day 7 expanded cells were used for all experiments. Notably, we observe no differences in phenotype, proliferative capacity and effector T cell function using mouse vs human recombinant IL-2.

### CD4⁺ CD25⁺ regulatory T cell extraction polarization and expansion

Spleens from wild type or PEG C57Bl/6 mice were excised and mashed with a 40 µm mesh nylon strainer, washed with PBS, and resuspended at a concentration of 1*10⁸ cells/mL in PBS supplemented with 2% FBS (VWR Seradigm cat #1500-500) and 1 mM EDTA (Sigma-Aldrich, cat #E7889). Tregs were extracted using Stemcell™ EasySep™ Mouse CD4+CD25+ Regulatory T Cell Isolation Kit II (Cat# 18763) and resuspended a concentration of 1*10⁶ cells/mL in complete DMEM media in a 24 well plate with 50µL/mL Dynabeads® Mouse T-Activator CD3/CD28 (Thermo Fisher Scientific Cat# 11456D) and 800 ng/mL human recombinant IL-2. After 48 h of incubation at 37 °C, 24 well plate was centrifuged at 300 g for 5 min and 500µL was gently removed and replaced with fresh media supplemented with IL-2. Cells were monitored and 1 ml of media replaced daily. At day 5 Dynabeads® were removed from T cells using Stemcell™ EasySep™ Magnet. Day 7 expanded cells were used for all experiments.

### Co-culture experiments

Day 6 BMMs were seeded a cell concentration of 0.25*10⁶ cells/well in a 24 plate in Complete RPMI containing β-Mercaptoethanol and 25 ng/mL MCSF and incubated overnight at 37 °C in a CO₂ incubator and allowed adhere to the bottom. In cases where BMMs were used for Immunohistochemistry, sterile glass slides were put at the bottom of the well for BMMs to adhere to. Wild-type or GFP *Leishmania major* parasites were added to culture at a preferred ratio and plates were incubated at 37 °C in a CO₂ incubator for 6 h. After incubation supernatant and non-penetrated parasites were removed. $0.25 \times 10^6$ cells/well of either control or PEG Th1 day 7 cells were added to each associated well for a further 24 h. In suppression co-culture experiments, prior to co-culture PEG Th1 cells were stained with either Celltracker Blue (CMAC; 20 µM) or Celltracker Green (CMFDA; 0.5 µM) to distinguish between Th1 cells and Tregs. Tregs were then added at a preferred ratio. For some experiments, 1 µL/mL of anti-IL-10 antibody (Biolegend) was added. Cells were allowed to incubate for 18 h and for experiments evaluating cytokine production, 1 µL/mL of Brefeldin A Solution (1000x) (Biolegend Cat# 420601) was added and incubated for additional 4 h. Cells were then stained from flow cytometry.

### ELISA analysis

ELISA supernatant analysis was performed with Eve Technologies Mouse Cytokine Proinflammatory Focused 10-Plex Discovery Assay® Array (MDF10). After normalizing for background cytokine expression, all expression values were expressed as % of maximal responses which was Th1 + *L. major* infected macrophage co-cultures. Expression values were then converted to log2 to describe increase or decrease fold change between the indicated experimental conditions. Values from three independent experiments are shown. Formula used: Log₂(experimental value/(average maximal response − average background response).

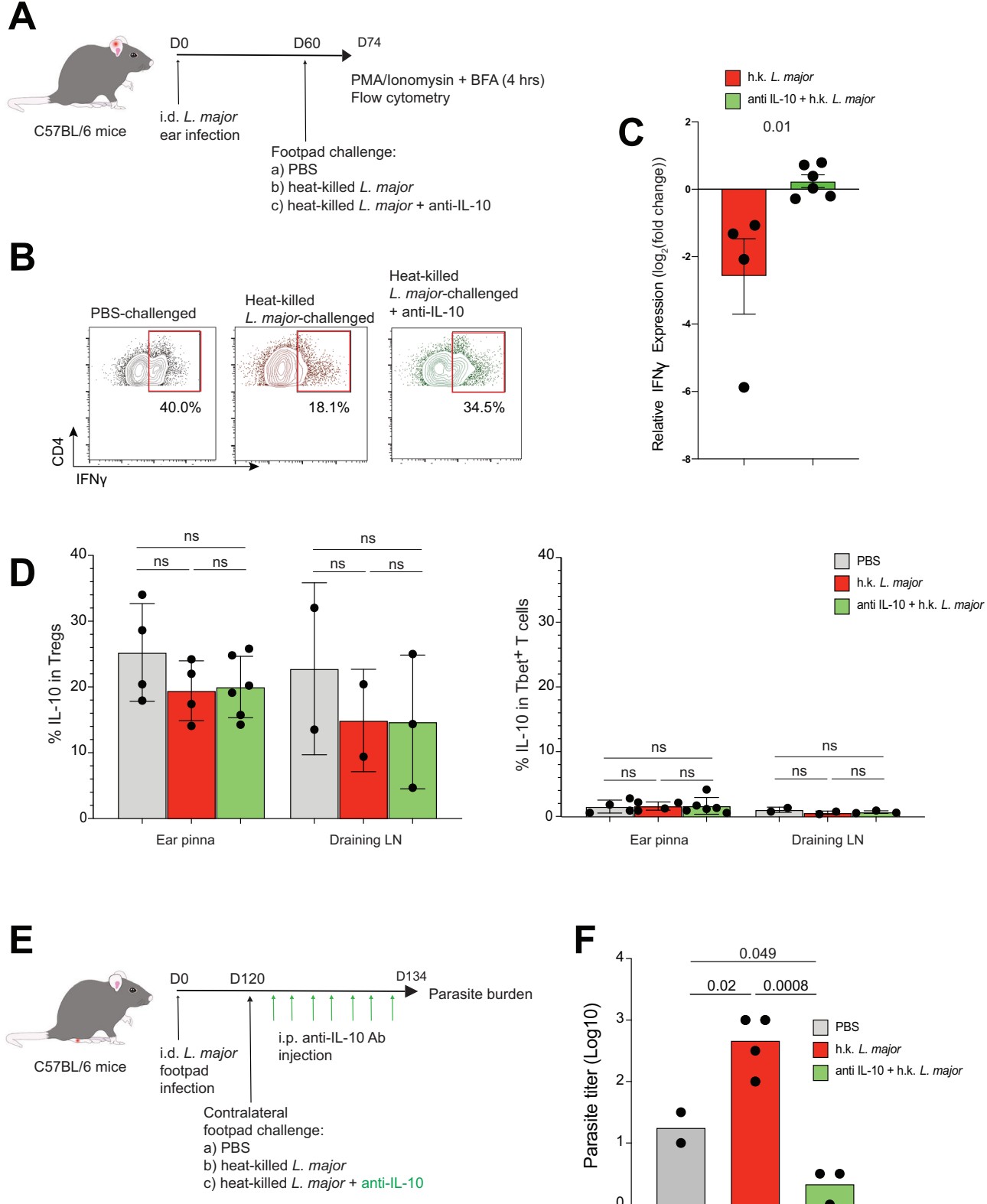

## Immunohistochemistry

After co-culture, T cells were gently washed off and BMMs were fixed with 2.5% Paraformaldehyde. Slides were washed, blocked with Fc blocker (Innovex), 4% mouse serum (ImmunoReagents) and 4% goat serum. The primary antibody used was rat anti-F4/80 (Abcam cat#ab6640) at 1:500 dilution. Secondary antibodies used were

AF568-conjugated goat anti-rat (Abcam cat#ab175476) at 1:1000 dilution, and AF488 conjugated chicken anti-GFP (Abcam cat#ab13970) at 1:5000 dilution. Slides were stained with Hoechst 33342 (Molecular Probes) for 30 min at 1:2000 dilution and mounted with ProLong Gold (Invitrogen). Images were acquired using the Zeiss AxioObserver confocal microscope and analyzed using ImageJ.

**Fig. 6 | Treg-mediated suppression of anti-*L. major* responses is driven by IL-10.** **A** C57Bl/6 mice were infected via intradermal ear injections with 1 million *L. major* parasites. 60 days post infection, mice were challenged with PBS or $5 \times 10^6$ heat-killed wild-type parasites into the footpad. 2 days prior to analysis, some mice challenged with wild-type heat-killed *L. major* were injected intra-peritoneally with anti-IL-10 antibody. The ears were prepared for flow cytometry analysis 14 days post challenge. **B**, **C** Percent and relative expression of IFNγ production compared to mice challenged with PBS in $CD45^+CD3^+CD4^+$ Tbet$^+$ T cells in the ears and draining lymph nodes. **B** Red box: positive IFNγ signal. **C** Red bar: IFNγ production by Th1 cells form mice challenged with heat-killed wild-type parasites, green bar: treated with anti-IL-10 antibody. Each dot represents a single ear, $n = 2$ h.k. *L. major* challenge, and 3 for h.k. PEPCK$^{-/-}$ *L. major* challenge. Two-tailed unpaired Student's *t* test. Mean +/− SD. **D** Percent IL-10 production in $CD45^+CD3^+CD4^+Foxp3^+$ and $CD45^+CD3^+CD4^+$ Tbet$^+$ cells in lesion ears and draining lymph nodes. Each dot represents a single ear or draining lymph node, $n = 2$ PBS or h.k. *L. major* challenge, and 3 for h.k. PEPCK$^{-/-}$ *L. major* challenge. Each dot represents a single ear or draining lymph node. Two-tailed unpaired Student's *t* test, ns not significant. Mean +/− SD. **E** C57Bl/6 mice were infected via footpad injections with 1 million *L. major* parasites. 120 days post infection, mice were challenged with PBS or $5 \times 10^6$ heat-killed wild-type parasites into the contralateral footpad. Some mice challenged with heat-killed wild-type *L. major* were injected intraperitoneally with anti-IL-10 antibody every 2 days over 2 weeks. The ears were prepared for parasite burden analysis 14 days post challenge. **F** Parasite burden from footpads of healed mice. Each dot represents parasite counts from each footpad ($n = 1$ experiment). Two-tailed unpaired Student's *t* test. Source data are provided as a Source Data file.

## Flow cytometry

At indicated times, mice were sacrificed and the ears, cervical dLNs, and inguinal non-dLNS were made into single-cell suspensions in complete RPMI medium. Ears were digested using LiberaseTM TL (Sigma, Cat# 5401020001) for 90 min at 37 °C. The cells were directly stained ex vivo for surface expression of CD4 (clone GK1.5), CD45 (clone 103114), CD3 (17A2 and CD25 (clone 3C7) (all from Biolegend) and intracellularly for Foxp3 (clone 3G3) or Tbet (clone 4B10) by using an eBioscience™ Foxp3 / Transcription Factor Staining Buffer Set (Cat # 00-5523-00). In some experiments, the cells were stimulated with Cell activation cocktail with Brefeldin A (BioLegend Cat #423304) and incubated at 37 °C for 4 h prior to flow cytometry staining for IL-10 and/or IFNγ. Phenotypic characterization of BMMs and CD4+T cells were performed using the BD FACSCanto-II and analyzed with FlowJo software (Tree Star). BMMs and T cells were phenotyped with a panel of directly conjugated anti-mouse mAb as indicated in the Key Resources Table.

## Live-cell imaging in 3D collagen chambers

Collagen type I was used to recapitulate the three-dimensional (3D) fibrillar networks[36,69]. Bovine collagen (PureCol) was used to achieve a final concentration of 1.7 mg/mL in each chamber. Cells were labeled with either Celltracker Blue (CMAC; 20 μM), Celltracker Orange (CMTMR; 5 μM) Celltracker Green (CMFDA; 0.5 μM) or Celltracker Deep Red (1 μM), washed and embedded into collagen. Chambers were allowed to solidify for 45 min at 37 °C/5% CO2 and placed onto a custom-made heating platform attached to a temperature controller apparatus (Werner Instruments). A thermocouple device was used to continuously monitor and maintain the chamber temperature at 37 °C. A multiphoton microscope with two Ti:sapphire lasers (Coherent) was tuned to between 780 and 920 nm for optimized excitation of the fluorescent probes used. For four-dimensional recordings of cell migration, stacks of 13 optical sections (512 ×512 pixels) with 4 μm z-spacing were acquired every 15 s to provide imaging volumes of 48 μm in depth. Emitted light was detected through 460/50 nm, 525/70 nm and 595/50 nm dichroic filters with non-descanned detectors. All images were acquired using the 20 × 1.0 N.A. Olympus objective lens (XLUMPLFLN; 2.0 mm WD).

## Adoptive transfer experiments and intravital microscopy

$10–15 \times 10^6$ day 7 control or PEG Th1 were stained with Celltracker Blue (CMAC; 20 μM), Celltracker Orange (CMTMR; 5 μM) or Celltracker Deep Red (1 μM), colors alternating between experiments[36]. Cells were washed 3 times with PBS and transferred into infected mice via intravenous injection. Mice were anaesthetized and infected ear was prepared for microscopy[70]. A multiphoton microscope with two Ti-sapphire lasers (Coherent) was tuned to between 780 and 920 nm for optimized excitation of the fluorescent probes used. For four-dimensional recordings of cell migration, stacks of 12 optical sections (512 × 512 pixels) with 4 μm z-spacing were acquired every 15 s to provide imaging volumes of 48 μm in depth. Emitted light was

detected through 460/50 nm, 525/70 nm and 595/50 nm dichroic filters with non-descanned detectors. All images were acquired using the 20 × 1.0 N.A. Olympus objective lens (XLUMPLFLN; 2.0 mm WD). Automated 3D tracking of T cell centroids was performed for cell motility analyses. Further cell track parameters (arrest coefficient and mean displacement) were analyzed in Matlab (Mathworks).

## Image analysis

Time lapse micrograph images were transformed using Imaris 8.3 (Bitplane) to generate maximum intensity projections (MIPs) and exported as Quicktime movies. Automated 3D tracking of T cell centroids was performed for all motility analyses, and subsequent cell track parameters (mean velocity 3D, arrest coefficient, mean displacement) were analyzed using a custom script in Matlab (Mathworks).

## Quantification and statistical analysis

Unpaired Student's *t* test and Mann−Whitney U test were used for comparisons of datasets with normal and non-normal distribution, respectively, using Prism 9 (GraphPad). Median and *p* values from statistical analyses are indicated in each graph. When *p* values were higher than 0.05, differences were considered as not significant.

## Reporting summary

Further information on research design is available in the Nature Portfolio Reporting Summary linked to this article.

## Data availability

The source data for each graph (Main and supplementary figures) are provided as a Source data file. Any additional inquiries should be addressed to Thomas T. Murooka or Jude E. Uzonna (corresponding authors). Source data are provided with this paper.

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

## Acknowledgements

We would like to thank Dr. Christine Zhang from the Flow core facility at the University of Manitoba. This work was supported in by the Canadian Institute for Health Research (CIHR) grants #148701 and #178224 to J.E. and Research Manitoba grant #44693 to T.T.M. R.Z. is a recipient of a graduate scholarship from Research Manitoba.

## Author contributions

R.Z., Z.M., J.U. and T.T.M. conceived and designed the experiments; R.Z., Z.M., A.Y., G.G., P.L., D.N. and W.K. performed the experiments; R.Z., Z.M., J.U. and T.T.M. analyzed the data, R.Z. and T.T.M. wrote the manuscript.

## Competing interests

The authors declare no competing interests.
