## [Peer Review File · Nature Communications]

Antigen recognition reinforces regulatory T cell mediated *Leishmania major* persistenceREVIEWER COMMENTS

Reviewer #1 (Remarks to the Author):

This manuscript address an important knowledge gap about Leishmania parasite persistence and the lack of sterile cure, even when lesions have resolved. This work uses innovative tools like cells from PEG mice to study Leishmania-specific T cell responses which has been an ongoing need for the field. Additionally, the antigen (Ag)-specific adoptive transfer competition experiments in vivo are an elegant approach to uncouple the influence of tissue architecture/microenvironment (like the collagen network) from T cell migration/speed. However, the limitation of this approach is the parasite burden is not uncoupled from the tissue microenvironment, although this reviewer acknowledges this is a major technical challenge. The quality and rigor of the data is appropriate using cutting-edge multiphoton imaging and 3D collagen matrixes coupled with tools to measure Ag-specific T cell responses, and the data supports the conclusions. The results are highly significant and provide novel insight into the mechanisms of parasite persistence driven by the host response which have not been described in such detail due to limitations of the tools available, until now. In general, the manuscript was easy to understand, and the scientific data generated using some very technical approaches was written so that is was still clear and concise.

The key result of the manuscript is that Ag-specific Tregs decrease Th1 effector responses better than control Tregs during *L. major* infection. Previous hypotheses propose parasites persist in safe haven cells like M2 macrophages in the skin, but these findings provide evidence for an alternative hypothesis, although not mutually exclusive, showing dermal Ag-specific Tregs regulate Th1 responses, thereby influencing parasite numbers. In brief, the parasites maintain an Ag-specific Treg population in the skin that further promotes the persistence of the parasites. However, missing piece of data to complete the story is the parasite burdens, especially for Figures 3, 5, and 6. The authors suggest larger lesions in Fig. 5D are due to parasite numbers (lines 256-257) but parasite burdens are not reported. This is especially important given parasite numbers do not always correlate with disease in cutaneous leishmaniasis and host-mediated immunopathology can contribute to lesions. For instance, please see work of Glennie et al. (PMID 26216123 and 28419151) where the inflammatory response (i.e. monocytes, DTH) in mice previously infected and healed did not correlate with parasite burdens upon challenge in the contralateral ear. While Figures 3, 5 and 6 address an important question in T cell biology, it would be more impactful if parasite burdens were reported (alongside 3B-D, 5D and 6B) to know if Treg IL-10 mediated regulation of CD4 effector cells equates to a functional consequence (i.e. parasite control).

Fig. 5E, F and G need to be corrected as they presently do not match results section text

Fig. 1F: data in results section text says results are significant, but no stats in figure and no error bars. This is important as it is one of the only figures quantifying parasites which adds significance to the manuscript showing PEG Th1 have a functional consequence on parasite burdens

For data like Fig. 3E showing IFN γ expression, these are challenging for the reader to determine how the analysis was performed by reading the figure legend. If it is based on flow cytometry from Fig. 3D using a % of the grey bar in some way as the legend suggests, the fold changes are not consistent between figures. If it is not from flow gating on Th1 cells, it is not clear if IFN γ was measured from all T cells in culture or if the PEG Th1 cells were sorted out for real time PCR or ELISA. Please clarify 3E, 3G, 6C, etc and 3I.

Fig. 3D states in legend unpaired ttest was used to assess statistical significance but the overlap between the range of data sets is surprising. To confirm, was this a paired ttest? Or a different statistical test? Or were all individual data in multiple experiments (rather than the average within a single experiment) used for analysis?

Minor:

Lines 106-109/Fig 1A-D and Supp Fig 1: Please clarify timing. From the methods section, it seems the parasites are given to macrophages prior to co-culture with T cells, and parasites were not given simultaneously as T cells and macrophages all cultured together, but this is not clear in the text of the results section. Please clarify. Also are any T cells harboring parasites in the co-cultures?

What restrains the control Th1 cells at 10-15 weeks p.i. when there are very few parasite numbers (Figure 2F)? While the data is robust and shows Ag-specificity contributes to decreasing velocity, but this decrease occurs in Ag-specific and Ag non-specific cells, decreasing the importance of Ag specificity which is a major theme of the findings. Please discuss.

What factor from T cell macrophage co-cultures (T cell or macrophage derived?) promotes CD40 and MHCII upregulation on macrophages?

Define hr and rh in hrIL-2 (example line 185) and rhIL-10 (example line 204) as typically this suggests recombinant human cytokine but not sure why you would use recombinant human cytokines for mouse assays.

Several flow cytometry figures have low resolution and poor image quality. Fig. 4J and Fig. 5E

Fig. 1D legend suggests uninfected or infected macrophages used but results sections text suggests only infected macrophages were used in assay. Please clarify

Fig. 1C legend: line 812 about red box might belong with Fig. 1A and B, not 1C

Specific number of transferred cells. It is not clear if 10-15 million stained cells were all transferred to a single animal as written

Typos: line 420 drRed instead of dsRed; line 439 maybe supposed to be hours instead of days; Fig. 6B-C legend to instead of 'too'; B6 and others C57BL/6, respectively instead of 'respectfully' several times

Note to reviewer: technically assessing confinement ratios (Fig. 2E and Fig. 4F) is outside my area of expertise.

Reviewer #2 (Remarks to the Author):

Insufficient clearing of Leishmania parasites during primary infections results in skin tissue healing as well as parasites persistence in tissue-resident macrophages. Zayats and colleagues hypothesised that parasite-specific Treg cells played a major role in immune escape and long-term disease. This study utilises an inhouse mouse model of transgenic CD4+ T cells recognising the Leishmania major immunodominant peptide PEPCK, called PEG T cells. In vitro studies using in vitro generated PEG CD4+ Th1 effector cells and PEG Treg cells clearly showed the importance of PEG Treg cells in neutralising Th1 cell-mediated Leishmania control. In vivo studies with adoptively transferred PEG Th1/Treg cells and/or polyclonal (PEG-unrelated) Th1/Treg cells they confirmed the importance of Treg-derived IL-10 in inhibiting the clearing of macrophage intracellular Leishmania species upon reactivation in "healed" mice. This is an interesting study, well carried out and discussed. Perhaps the new model will help to clarify the mechanism underlying Treg cell generation in the first place, during primary Leishmania infections. I wish to discuss the following points.

1) Adoptively transferred PEG Th1 cells in "healed" mice did not home properly, as suggested from the ear tissue staining in Fig 2B. Either this is due to suboptimal data processing or the consequence of Treg-mediated inhibition of Th1 cell recruitment. Please clarify.

2) The study emphasises the antigen-driven Treg suppression of anti-Leishmania Th1 responses. But Fig 3E-G demonstrate that even polyclonal Treg cells give substantial inhibitory responses. Please explain and discuss.

3) Based on the IHC data shown in Fig 4B, I conclude that addition of PEG Tregs induced a switch from PEG Th1 - macrophage interactions to substantial PEG Th1 - PEG Treg interactions. Am I wrong?

4) I do not understand the labelling in Fig 4C.

5) The data in Fig 5B are very interesting and clearly demonstrate Leishmania accumulation and Treg cell expansion. Out of curiosity, do you think Treg cells expand locally or in draining LNs (prior to their tissue homing)?

Reviewer #3 (Remarks to the Author):

Zayats and collaborators used an experimental mouse model of cutaneous leishmaniasis to investigate the role of regulatory T cells (Tregs) in the persistence of the protozoan parasite *Leishmania major* in the skin. The authors previously identified a peptide derived from PEPCK as an immunodominant peptide that elicit a dominant CD4+ T cell response in mice and humans infected with *Leishmania*. They generated PEPCK-specific TCR transgenic mice where CD4+ T cells recognize the PEPCK peptide in the context of I-Ab (PEG mice). Using advance two-photon microscopy, they have observed and described PEG Th1:macrophage interactions during *L. major* infection. Furthermore, they demonstrated that *Leishmania*-specific Tregs are highly suppressive through the production of IL-10 and that expansion of those endogenous Tregs play an important role in the reactivation of diseases at the primary infection site. This study has been carefully executed, is well-controlled, data is clearly presented, and the conclusions are fully supported by the data. There are a number of issues to be addressed to further strengthen the conclusions of this study.

1- It is unclear whether the authors infected mice with stationnary phase or metacyclic promastigotes. If they used metacyclics, the enrichment method must be described in the Methods section.

2- Mouse infection were performed through intra-dermal inoculation of 1×10^6 promastigotes. This is a high dose of parasites, considering that between 100 and 10^4 parasites are inoculated during natural transmission. What was the rationale for using this high parasite dose? The authors previously reported that inoculation of a high dose of parasites predominantly induced proliferation and cytokine production by CD4+ T cells. By contrast, low dose infection was dominated by CD8+ T cells. Is it possible that infecting PEG mice with a low dose of metacyclic *L. major* promastigotes would lead to a different outcome with respect to *Leishmania*-specific Treg expansion and activity? Can this be experimentally addressed?

3- This study was performed using male mice. Whether the reported findings apply to female mice remains to be determined.

Response to reviewer's comments

We have addressed all concerns raised by the reviewers in a point-by-point response indicated below. The attached revised manuscript also incorporates new data to further support our overall findings and all revised text is highlighted in red.

Response to Reviewer #1

- 1) *This manuscript address an important knowledge gap about Leishmania parasite persistence and the lack of sterile cure, even when lesions have resolved. This work uses innovative tools like cells from PEG mice to study Leishmania-specific T cell responses which has been an ongoing need for the field. Additionally, the antigen (Ag)-specific adoptive transfer competition experiments in vivo are an elegant approach to uncouple the influence of tissue architecture/microenvironment (like the collagen network) from T cell migration/speed. However, the limitation of this approach is the parasite burden is not uncoupled from the tissue microenvironment, although this reviewer acknowledges this is a major technical challenge. The quality and rigor of the data is appropriate using cutting-edge multiphoton imaging and 3D collagen matrixes coupled with tools to measure Ag-specific T cell responses, and the data supports the conclusions. The results are highly significant and provide novel insight into the mechanisms of parasite persistence driven by the host response which have not been described in such detail due to limitations of the tools available, until now. In general, the manuscript was easy to understand, and the scientific data generated using some very technical approaches was written so that is was still clear and concise.*

Authors response: We thank this reviewer for the positive comments.

- 2) *The key result of the manuscript is that Ag-specific Tregs decrease Th1 effector responses better than control Tregs during L. major infection. Previous hypotheses propose parasites persist in safe haven cells like M2 macrophages in the skin, but these findings provide evidence for an alternative hypothesis, although not mutually exclusive, showing dermal Ag-specific Tregs regulate Th1 responses, thereby influencing parasite numbers. In brief, the parasites maintain an Ag-specific Treg population in the skin that further promotes the persistence of the parasites. However, missing piece of data to complete the story is the parasite burdens, especially for Figures 3, 5, and 6. The authors suggest larger lesions in Fig. 5D are due to parasite numbers (lines 256-257) but parasite burdens are not reported. This is especially important given parasite numbers do not always correlate with disease in cutaneous leishmaniasis and host-mediated immunopathology can contribute to lesions. For instance, please see work of Glennie et al. (PMID 26216123 and 28419151) where the inflammatory response (i.e. monocytes, DTH) in mice previously infected and healed did not correlate with parasite burdens upon challenge in the contralateral ear. While Figures 3, 5 and 6 address an important question in T cell biology, it would be more impactful if parasite burdens were reported (alongside 3B-D, 5D and 6B) to know if Treg IL-10 mediated regulation of CD4 effector cells equates to a functional consequence (i.e. parasite control).*

Authors response: We appreciate the reviewer's insightful feedback, and we agree that measuring parasite burden in Figures 3, 5, and 6 is an important readout. We have conducted and included new data to address comments in the revised manuscript, described in detail below:

We now include two independent *in vitro* parasite burden experiments in revised Figure 3, where GFP-*L. major*-infected BMMs were cultured alone, with PEG Th1 cells alone, PEG Th1 cells + control Tregs, or PEG Th1 cells + PEG Tregs. After 24 hours of culture, T cells were gently washed off and BMMs were stained for macrophage marker F4/80, GFP to enhance the brightness of parasites, and Hoechst nuclear dye (Lines 210-218). We observed clear reduction in parasite killing capacity of activated macrophages when Tregs were present in culture *in vitro*. Antigen recognition by PEG Tregs enhanced suppression and led to a higher number of surviving parasites, consistent with the overall conclusions of this study (Figure 3I-K).

In the revised Figure 5E, we now include new data on tissue parasite burden after challenge with the indicated parasites (Lines 268-269, Figure 5E). C57BL/6 mice were infected with 1×10^6 stationary phase *L. major* parasites intradermally into the ear pinna and allowed to heal for 60 days. Some mice were then challenged with PBS into the footpad, while others received 5×10^6 heat-killed wildtype *L. major* promastigotes or 5×10^6 heat-killed PEPCK-deficient *L. major* promastigotes. Parasite burden was determined two weeks later using the serial dilution technique and show that parasite burden largely correlated with immunohistochemistry studies and lesion score. When mice were challenged with heat-killed wildtype *L. major*, parasite burden was increased 100-fold after 2 weeks, consistent with increased numbers of dsRed⁺ *L. major*, whereas and challenge with heat-killed PEPCK-deficient *L. major* resulted in significantly lower parasite burden in ears. These studies reinforce *in vitro* studies that antigen activate Treg functions *in vivo*.

Lastly, revised Fig 6 now includes tissue parasite burden data after *in vivo* IL-10 blockade. Specifically, C57BL/6 mice were infected with 1×10^6 stationary phase *L. major* parasites into the footpad and after 120 days, challenged with either PBS or 5×10^6 heat-killed wildtype *L. major* promastigotes with or without anti-IL-10 antibody blockade every 2 days over 2 weeks (Lines 284-294, Figure 6E, F). We observed a striking reduction in parasite burden upon IL-10 blockade treatment even in the presence of expanded Tregs. These findings further extend studies by Belkaid et al. (PMID 11714756) by showing that mice deficient in IL-10 can achieve sterile cure, possibly by blunting Treg effector function.

3) *Fig. 5E, F and G need to be corrected as they presently do not match results section text*

Authors response: We thank the reviewer for noting this error. We have fixed the figure labeling and sections in the text that also includes new data (Lines 269-274).

4) *Fig. 1F: data in results section text says results are significant, but no stats in figure and no error bars. This is important as it is one of the only figures quantifying parasites which adds significance to the manuscript showing PEG Th1 have a functional consequence on parasite burdens*

Authors response: To improve quantitative power, we performed two additional independent studies that is now included in this revised Figure 1F. Description of statistical analysis is included

in the figure legends. In all experiments, PEG Th1 exerts strong immune responses to infection by significantly reducing parasite burden in macrophages.

5) *For data like Fig. 3E showing IFN γ expression, these are challenging for the reader to determine how the analysis was performed by reading the figure legend. If it is based on flow cytometry from Fig. 3D using a % of the grey bar in some way as the legend suggests, the fold changes are not consistent between figures. If it is not from flow gating on Th1 cells, it is not clear if IFN γ was measured from all T cells in culture or if the PEG Th1 cells were sorted out for real time PCR or ELISA. Please clarify 3E, 3G, 6C, etc and 3I.*

Authors response: Data presented as Log₂ fold change better represents the reduction in cytokine production and is often used to represent fold increase/decreased in gene expression in transcriptomics analyses. Data in Fig. 3E was calculated as follows: after normalizing for background cytokine expression, all expression values were expressed as % of maximal responses which was Th1 + *L. major* infected macrophage co-cultures. Expression values were then converted to log₂ to describe increase or decrease fold change between the indicated experimental conditions. Values from three independent experiments are shown. Formula: Log₂(experimental value/(average maximal response – average background response)). This description has been included in the methods section (lines 521-526).

6) *Fig. 3D states in legend unpaired ttest was used to assess statistical significance but the overlap between the range of data sets is surprising. To confirm, was this a paired ttest? Or a different statistical test? Or were all individual data in multiple experiments (rather than the average within a single experiment) used for analysis?*

Authors response: We thank this reviewer for catching this mistake: previous statistical analysis was mistakenly applied to an individual data set from a single experiment. The revised figure depicts mean values across three independent experiments and the correct statistical analysis. We now show that statistically significant reduction in IFN γ responses is observed primarily at the 1:1 effector:Treg ratio.

7) *Minor:*

Lines 106-109/Fig 1A-D and Supp Fig 1: Please clarify timing. From the methods section, it seems the parasites are given to macrophages prior to co-culture with T cells, and parasites were not given simultaneously as T cells and macrophages all cultured together, but this is not clear in the text of the results section. Please clarify. Also are any T cells harboring parasites in the co-cultures?

Authors response: The reviewer's understanding of our methods is correct, and we have clarified the timing of these experiments in the revised text (line 110). Parasites were given to macrophages and allowed to adequately infect their target cells for 6 hours. Afterwards, the non-internalized parasites were washed off and T cell populations were added. In our experiments, we did not observe T cells harboring parasites: *L. major* was found exclusively within macrophages.

8) *What restrains the control Th1 cells at 10-15 weeks p.i. when there are very few parasite numbers (Figure 2F)? While the data is robust and shows Ag-specificity contributes to decreasing velocity, but this decrease occurs in Ag-specific and Ag non-specific cells, decreasing the importance of Ag specificity which is a major theme of the findings. Please discuss.*

Authors response: Substantial structural changes in healed skin are often observed by 2P microscopy of skin: the collagen layer of the ear pinna is less uniform, and pockets of residually infected cells are typically associated with dysregulated collagen networks in these studies (data not shown). Such changes to the ECM and how they can potentially impact T cell migratory behaviors, regardless of antigen specificity, is discussed in more detail (lines 325-341). In addition, substantial immunological alterations induced by the presence of *L. major*-specific Tregs may further impact T cell recruitment and motility responses by suppressing expression of inflammation-driven chemotactic and tissue remodeling cues. In mice, Tregs were shown to accumulate in skin after wounding and that their deletion resulted in IFN γ -dependent accumulation of pro-inflammatory macrophages and attenuated wound closure (PMID: 26826250). In lungs of mice, Tregs have been shown to preferentially mediate tissue repair as well (PMID: 26317471). Together, structural and immunological changes during chronic *L. major* infection in skin is likely to impose restrictions on general T cell recruitment and motility behaviors. However, even within these tissue confines, we still observe a clear role of antigen in regulating effector T cell migratory behaviors at 10-15 weeks post-infection, implicating their importance in ongoing parasite control.

9) *What factor from T cell macrophage co-cultures (T cell or macrophage derived?) promotes CD40 and MHCII upregulation on macrophages?*

Authors response: Macrophages constitutively express CD40 and MHC II at low levels. IFN γ -induced CD40 expression involves activation of STAT-1 α and NF κ B activation through an autocrine response to IFN γ -mediated TNF production. CD40-CD40L binding induces further upregulation of MHC II, CD40, CD80 and CD86, creating a positive amplification loop for further macrophage activation (PMID: 16020513).

10) *Define hr and rh in hrIL-2 (example line 185) and rhIL-10 (example line 204) as typically this suggests recombinant human cytokine but not sure why you would use recombinant human cytokines for mouse assays.*

Authors response: “rh” refers to ‘recombinant human’. Early pilot experiments compared murine T cell proliferation, phenotype and function after expansion in either mouse vs human recombinant IL-2 and found no differences in all these readouts. Similar observations were made with our Treg cultures, and for these reasons, we continued to use human recombinant IL-2 (rhIL-2) for the majority of our studies. To improve clarify, we have removed the abbreviation “rh” from the main text and inserted a sentence in the methods to indicate the use of human IL-2 (lines 485-485).

11) *Several flow cytometry figures have low resolution and poor image quality. Fig. 4J and Fig. 5E*

Authors response: We have replaced figures 4J (Now 4N) and 5E (Now 5F) with higher resolution counterparts.

12) *Fig. 1D legend suggests uninfected or infected macrophages used but results sections text suggests only infected macrophages were used in assay. Please clarify*

Authors response: We have corrected this error in the figure legend. Only infected macrophages were used in these studies (line 838).

13) *Fig. 1C legend: line 812 about red box might belong with Fig. 1A and B, not 1C*

Authors response: We have corrected this error.

14) *Specific number of transferred cells. It is not clear if 10-15 million stained cells were all transferred to a single animal as written*

Authors response: Each animal received 10-15 million control and 10-15 million PEG Th1 cells. We have clarified the figure legend.

15) *Typos: line 420 drRed instead of dsRed; line 439 maybe supposed to be hours instead of days; Fig. 6B-C legend to instead of 'too'; B6 and others C57BL/6, respectively instead of 'respectfully' several times*

Authors response: These errors have all been corrected in the revised manuscript.

Response to Reviewer #2

16) *Insufficient clearing of Leishmania parasites during primary infections results in skin tissue healing as well as parasites persistence in tissue-resident macrophages. Zayats and colleagues hypothesised that parasite-specific Treg cells played a major role in immune escape and long-term disease. This study utilises an inhouse mouse model of transgenic CD4+ T cells recognising the Leishmania major immunodominant peptide PEPCK, called PEG T cells. In vitro studies using in vitro generated PEG CD4+ Th1 effector cells and PEG Treg cells clearly showed the importance of PEG Treg cells in neutralising Th1 cell-mediated Leishmania control. In vivo studies with adoptively transferred PEG Th1/Treg cells and/or polyclonal (PEG-unrelated) Th1/Treg cells they confirmed the importance of Treg-derived IL-10 in inhibiting the clearing of macrophage intracellular Leishmania species upon reactivation in "healed" mice. This is an interesting study, well carried out and discussed. Perhaps the new model will help to clarify the mechanism underlying Treg cell generation in the first place, during primary Leishmania infections. I wish to discuss the following points.*

Authors response: We thank this reviewer for the positive comments. We would like to point out that Tregs were not adoptively transferred into recipient mice, but rather were endogenously

expanded *in vivo*. In these studies, injection with heat-killed parasite led to expansion of an endogenous pool of antigen-specific Tregs, and the consequence of this expansion was measured using several readouts.

17) *Adoptively transferred PEG Th1 cells in “healed” mice did not home properly, as suggested from the ear tissue staining in Fig 2B. Either this is due to suboptimal data processing or the consequence of Treg-mediated inhibition of Th1 cell recruitment. Please clarify.*

Authors response: The reviewer is correct in that significantly reduced numbers of PEG Th1 cells are recruited into “healed” skin. In order to make sure sufficient experimental replicates and sampling was done, T cell motility analysis from Fig. 2B was pooled from 8 independent experiments, 13 total mice, and 2-3 movies per ear/mouse: we do not believe the data presented is due to suboptimal data processing.

As discussed in response to question #8 from Reviewer 1, substantial structural changes in healed skin are often observed by 2P microscopy of skin: the collagen layer of the ear pinna is less uniform and pockets of residually infected cells are typically associated with dysregulated collagen networks in these studies (data not shown). Such changes to the ECM and how they can potentially impact T cell migratory behaviors, regardless of antigen specificity, is discussed in more detail (lines 325-341). In addition, substantial immunological alterations induced by the presence of *L. major*-specific Tregs may further impact T cell recruitment and motility responses by suppressing inflammation-driven chemotactic and tissue remodeling cues. In mice, Tregs were shown to accumulate in skin after wounding and deletion of Tregs resulted in IFN γ -dependent accumulation of pro-inflammatory macrophages and attenuated wound closure (PMID: 26826250). In lungs of mice, Tregs have been shown to preferentially mediate tissue repair as well (PMID: 26317471). Together, structural and immunological changes during chronic *L. major* infection in skin is likely to impose restrictions on general T cell recruitment and motility features. However, even within these tissues we still observe a clear role of antigen in regulating effector T cell migratory behaviors at 10-15 weeks post-infection, implicating their importance in ongoing parasite control.

18) *The study emphasises the antigen-driven Treg suppression of anti-Leishmania Th1 responses. But Fig 3E-G demonstrate that even polyclonal Treg cells give substantial inhibitory responses. Please explain and discuss.*

Authors response: This is a good point raised by this reviewer. We indeed observe that non-specific Tregs do possess baseline suppressive activity, which is not surprising because they express some immunomodulatory cytokines such as IL-10 (Fig.3F) and moderate levels of immune checkpoint molecules including PD-1 and LAG-3 (Reviewer figure 1, blue bars). However, the important observation here is that antigenic stimulation greatly enhances Treg function through increased IL-10 production and expression of the aforementioned checkpoint molecules both before and after *in vivo* expansion (Reviewer figure 1, pink bars). This is consistent with several reports that while TCR stimulation is not required to maintain FoxP3 expression, it is critically

*Figure 1: PD-1, GITR and LAG-3 expression in specific and non-specific Tregs in skin. Mice were infected with *L. major* and left to heal for 8 weeks. Mice were subsequently challenged with either PBS or heat-killed wildtype *L. major* as described in Figure 5A. At 14 days, control and PEPCK-specific Tregs were identified using tetramer and phenotypic analysis by flow cytometry was performed. Each data point represents a single mouse in one representative study. hk =heat killed.*

important to maintain effector Treg homeostasis and suppressive function to prevent overt autoimmunity *in vivo* (PMID: 25263123, 28099866). In the context of chronic *L. major* lesions in skin, we argue that continual TCR stimulation reinforces high suppressive phenotypes by effector Tregs (high PD-1, GITR and LAG-3) seen in PEPCK-specific Tregs, and that they play a more important role in parasite persistence compared to non-PEPCK specific Tregs.

19) Based on the IHC data shown in Fig 4B, I conclude that addition of PEG Tregs induced a switch from PEG Th1 – macrophage interactions to substantial PEG Th1 – PEG Treg interactions. Am I wrong?

Authors response: To clarify, Fig 4B are representative micrographs of live cell imaging studies performed in 3D collagen matrices. While these images and movies do seem to suggest increased clustering of PEG Th1 – PEG Treg cells, this occurs exclusively in the proximity of infected macrophages. From all of our imaging studies, we rarely observed prolonged interactions between effector and regulatory T cells in the absence of macrophages. The large clusters of cells make it impossible to distinguish between T cell-macrophage and T cell-T cell interactions, and therefore we are not able to determine whether Tregs drives a shift in cell-cell contact dynamics. So while the reviewer is correct in that greater Th1:Treg interactions may occur through engagement with macrophages, strong conclusions about a possible shift in responses cannot be made from our imaging studies.

20) I do not understand the labelling in Fig 4C.

Authors response: Figure 4C has been revised, and now features data in two separate graphs for better clarification (Fig 4C, H). In the new Fig 4C, we first describe how *L. major* infection impacts T cell:macrophage contact durations between the two T cell populations. As expected, PEG T cells engaged in prolonged contacts with infected macrophages. Next, we used this imaging approach to address whether Tregs impacted Th1:macrophage contact duration (Fig 4H), where *Leishmania*-specific Tregs (pink box) did not disrupt prolonged Th1:macrophage contacts. Appropriate revisions in the text were made to reflect the new figures.

21) The data in Fig 5B are very interesting and clearly demonstrate *Leishmania* accumulation and Treg cell expansion. Out of curiosity, do you think Treg cells expand locally or in draining LNs (prior to their tissue homing)?

Authors response: The source of expanded Tregs in the lesion remains an outstanding question for future studies. CCR5^{-/-} mice displayed higher numbers of parasite-specific effector T cells at the site of infection and significantly lower frequency of IL-10-producing cells (PMID: 17015634), suggesting that CCR5 facilitates recruitment of IL-10⁺ Tregs to the lesion site (where CCR5 ligand expression is high). Our data in Figure 5G seems to support this notion, where increased numbers of PEPCK-specific Tregs in skin lesions correspond with their reduced numbers in the draining cervical lymph node. While these data do not provide information on directional movement between tissues, future studies will focus on chemokine receptors and ligands (such as CCR5) that are involved in this process. We also cannot rule out the contribution of local Treg expansion in skin, and for this Ki67 staining by IHC or Edu *in vivo* proliferation readouts can be used to detect recent cell division events within Treg populations in the lesion.

Response to Reviewer #3

22) *Zayats and collaborators used an experimental mouse model of cutaneous leishmaniasis to investigate the role of regulatory T cells (Tregs) in the persistence of the protozoan parasite Leishmania major in the skin. The authors previously identified a peptide derived from PEPCK as an immunodominant peptide that elicit a dominant CD4+ T cell response in mice and humans infected with Leishmania. They generated PEPCK-specific TCR transgenic mice where CD4+ T cells recognize the PEPCK peptide in the context of I-Ab (PEG mice). Using advance two-photon microscopy, they have observed and described PEG Th1:macrophage interactions during L. major infection. Furthermore, they demonstrated that Leishmania-specific Tregs are highly suppressive through the production of IL-10 and that expansion of those endogenous Tregs play an important role in the reactivation of diseases at the primary infection site. This study has been carefully executed, is well-controlled, data is clearly presented, and the conclusions are fully supported by the data. There are a number of issues to be addressed to further strengthen the conclusions of this study.*

Authors response: We thank this reviewer for the positive comments.

23) *It is unclear whether the authors infected mice with stationnary phase or metacyclic promastigotes. If they used metacyclics, the enrichment method must be described in the Methods section.*

Authors response: We confirm that stationary phase promastigotes were used in the study. We have clarified this in the text (Lines 139, 444).

24) *Mouse infection were performed through intra-dermal inoculation of 1X10*6 promastigotes. This is a high dose of parasites, considering that between 100 and 10*4 parasites are inoculated during natural transmission. What was the rationale for using this high parasite dose? The authors previously reported that inoculation of a high dose of parasites predominantly induced proliferation and cytokine production by CD4+ T cells. By contrast, low dose infection was dominated by CD8+ T cells. Is it possible that infecting PEG mice with a low dose of metacyclic L. major promastigotes would lead to a different outcome with*

respect to Leishmania-specific Treg expansion and activity? Can this be experimentally addressed?

Authors response: We thank this reviewer for this important question. Indeed, our previous work showed that high dose parasite challenge preferentially induced strong CD4⁺ Th1 responses that contributed to immunity against secondary parasite challenge. With the focus on our current study being the molecular regulation of Th1:Treg interactions during chronic stages of cutaneous *L. major* infection, we felt that infection at a high parasite dose was appropriate to generate robust Th1 (and likely Treg) responses *in vivo*. Another important rationale was due to technical challenges visualizing residual parasites in healed skin: we performed 2P imaging studies by titrating down infecting parasite dose to 10⁵ and found that while residual parasites were present at >15 weeks post-infection, they were more difficult to visualize consistently across multiple timepoints. Another issue was frequent photobleaching of GFP⁺ or DsRed⁺ parasites when parasite numbers/cell was low, which made time-lapse imaging challenging. Therefore, infection studies at 10⁶ *L. major* parasites was optimal from an immunological and technical standpoint to study the interplay between effector and regulatory T cells, and to improve our ability to visualize chronic lesions over time. We also argue that while up to 10⁴ parasites are inoculated during a sandfly bite, multiple bites can increase total *L. major* antigen availability, which needs to be taken into account when evaluating the expansion and contraction dynamics of effector and regulatory T cells.

Nevertheless, this question is worthy of future investigation. For example, data shows that vaccinating OT-II mice with OVA peptides at 1 µg to 50 µg doses leads to expansion of OVA-specific Tregs from 5% to 45% in the inguinal lymph nodes. However, a 100 µg peptide dose decreased Treg proportions to 24%, indicating that Treg expansion may be dose and context dependent (such as the adjuvant) (PMID: 29467445). The expansion dynamics of PEPCK-specific Tregs during low and high dose challenge, and their role in parasite persistence in healed skin is subject of our future studies, but beyond the scope of our current studies.

25) *This study was performed using male mice. Whether the reported findings apply to female mice remains to be determined.*

Authors response: We agree, the exclusive use of male mice does not allow us to make broad conclusions on T cell regulation across both sexes and represents a limitation of this study. Recent studies in airway allergic responses provide evidence for hormonal regulation of Treg numbers and function, highlighting the importance of biological sex, including hormone regulation, as an important variable during chronic disease processes (PMID: 32025767). Comparative studies in both male and female recipients will permit investigations into whether sex hormones (and other biological factors that are different in males vs females) regulate Treg function in healed *L. major* lesions, and is our current investigative focus.

REVIEWERS' COMMENTS

Reviewer #1 (Remarks to the Author):

The authors have addressed my concerns and put forth a notable effort in adding supporting experiments and amending the text. A couple minor mistakes that will need to be addressed prior to publication are as follows but otherwise, the manuscript is suitable for publication.

1. Line 905 for figure legend 3 lists the wrong letters (should not be F and G but rather J and K)
2. the number of times the experiment in Fig. 6F was completed should be added to the legend, even if it was only one time.

Reviewer #2 (Remarks to the Author):

N/A

Reviewer #3 (Remarks to the Author):

The authors have addressed my main concerns.

However, the fact that only male mice were used in the study should be reflected in the title, abstract, and throughout the text. In addition, the authors must state in the Discussion that this is a limitation of this study.

Response to reviewer's comments

Please find our responses to reviewer's comments. The attached revised manuscript incorporates suggestions brought by the reviewers and editorial staff, highlighted in red.

Reviewer #1:

The authors have addressed my concerns and put forth a notable effort in adding supporting experiments and amending the text. A couple minor mistakes that will need to be addressed prior to publication are as follows but otherwise, the manuscript is suitable for publication.

1. Line 905 for figure legend 3 lists the wrong letters (should not be F and G but rather J and K)

This has been corrected. Please note that one of the panels was moved to Suppl Figure 3, and therefore the panel numbering has been revised.

2. the number of times the experiment in Fig. 6F was completed should be added to the legend, even if it was only one time.

The number of times this experiment was performed has now been added to the legend.

Reviewer #3:

The authors have addressed my main concerns.

However, the fact that only male mice were used in the study should be reflected in the title, abstract, and throughout the text. In addition, the authors must state in the Discussion that this is a limitation of this study.

We agree, and have revised/stated in the abstract and discussion that the conclusions are based on studies exclusively in male mice, that thus a limitation of this work. Lines 404-408.